# NLRC5 inhibits neointima formation following vascular injury and directly interacts with PPARγ

Peipei Luan[1,2], Weixia Jian[2], Xu Xu[1], Wenxin Kou[1], Qing Yu[1], Handan Hu[3], Dali Li [3], Wei Wang[4], Mark W. Feinberg [5], Jianhui Zhuang[1], Yawei Xu[1] & Wenhui Peng [1]

NLR Family CARD Domain Containing 5 (NLRC5), an important immune regulator in innate immunity, is involved in regulating inflammation and antigen presentation. However, the role of NLRC5 in vascular remodeling remains unknown. Here we report the role of NLRC5 on vascular remodeling and provide a better understanding of its underlying mechanism. *Nlrc5* knockout (*Nlrc5*[−/−]) mice exhibit more severe intimal hyperplasia compared with wild-type mice after carotid ligation. Ex vivo data shows that NLRC5 deficiency leads to increased proliferation and migration of human aortic smooth muscle cells (HASMCs). NLRC5 binds to PPARγ and inhibits HASMC dedifferentiation. NACHT domain of NLRC5 is essential for the interaction with PPARγ and stimulation of PPARγ activity. Pioglitazone significantly rescues excessive intimal hyperplasia in *Nlrc5*[−/−] mice and attenuates the increased proliferation and dedifferentiation in *NLRC5-deficient* HASMCs. Our study demonstrates that NLRC5 regulates vascular remodeling by directly inhibiting SMC dysfunction via its interaction with PPARγ.

[1] Department of Cardiology, Shanghai Tenth People's Hospital, School of Medicine, Tongji University, Shanghai 200072, China. [2] Department of Endocrinology, Xinhua Hospital, Shanghai Jiaotong University, School of Medicine, Shanghai 200092, China. [3] Shanghai Key Laboratory of Regulatory Biology, Institute of Biomedical Sciences and School of Life Sciences, East China Normal University, Shanghai 200241, China. [4] Columbia Center for Translational Immunology, Columbia University Medical Center, New York 10032, USA. [5] Cardiovascular Division, Department of Medicine, Brigham and Women's Hospital, Harvard Medical School, Boston, MA 02115, USA. Correspondence and requests for materials should be addressed to J.Z. (email: jh_zhuang@tongji.edu.cn) or to Y.X. (email: xuyawei@tongji.edu.cn) or to W.P. (email: pwenhui@tongji.edu.cn)

In response to injury, vascular smooth muscle cells (VSMCs) migrate and proliferate from the media into the intima. Such process is called neointima formation or neointima hyperplasia, which leads to vascular remodeling followed by potential atherosclerosis progression, in-stent restenosis or vein bypass graft failure[1,2]. Previous studies have shown that activation of both the innate and adaptive immune systems are involved in the pathogenesis of neointima hyperplasia and vascular remodeling[3,4]. However, medical therapies for inhibiting intima hyperplasia are limited. This is largely because mechanisms through which these immune modulators regulate vascular remodeling are poorly understood.

Recent evidence highlights the importance of specific innate immunity signaling pathways activated in vascular dysfunction and repair[5–7]. Innate immunity distinguishes a diversified set of extracellular and intracellular danger signals that primarily originate from microbes by groups of pattern recognition receptors including Toll-like receptors (TLRs) and Nod-like receptors (NLRs)[8,9]. NLRs are a group of evolutionarily conserved intracellular pattern recognition receptors that are useful in the detection of microbes and danger signals, and play a vital role in innate immunity and host physiology[10]. Notably, mutations or single nucleotide polymorphisms in these genes associate with human diseases including auto-immune disease, gastric cancer, early-onset menopause, among others[11–13].

Among multiple members of NLRs family, NOD-like receptor family CARD domain containing 5 (NLRC5) has been reported to be critical in antigen presentation, inflammation, and tissue fibrosis[14]. NLRC5 is abundantly expressed in immune cells in spleen, lymph node, and bone marrow[15]. NLRC5 is also highly expressed in lung and intestine, suggesting that the functions of NLRC5 are not limited to pathogen recognition[15,16]. NLRC5 shuttles between the cytoplasm and the nucleus in a cytokine response modifier A-dependent manner and acts as a key regulator of major histocompatibility complex (MHC) class I-dependent immune responses by cooperating with regulatory factor X5 (RFX5)[17–19]. In particular, it negatively regulates the NF-κB signaling, type I interferon activities, and the JAK2/STAT3-signaling pathway[20,21]. Likewise, our group has recently found that NLRC5 deficiency ameliorates diabetic nephropathy (DN) by alleviating chronic inflammation[22].

In this study, we demonstrate the protective role of NLRC5 in intimal hyperplasia, and the suppressive effect of NLRC5 on proliferation, migration, and dedifferentiation of VSMCs. Furthermore, we show a mechanistic link between NLRC5 and PPARγ.

## Results

**NLRC5 increased during vascular remodeling.** Vascular remodeling is considered a major feature in vascular diseases, such as in-stent restenosis, Kawasaki disease, and atherosclerosis[23]. Therefore, we first examined the expression of NLRC5 in coronary arteries from patients with Kawasaki disease and in coronary plaques from patients undergoing coronary artery bypass graft surgery. Compared with normal coronary arteries, NLRC5 expression was more abundant in VSMCs in both Kawasaki disease (Supplementary Fig. 1) and coronary plaques; however, its expression was also ubiquitous, rather than localized in the proliferative medial layer (Fig. 1a–c and Supplementary Fig. 2). We further investigated the expression of Nlrc5 in a vascular injury model of complete carotid ligation. Similar to what we observed in human diseased arteries, Nlrc5 expression dramatically increased in carotid arteries after injury (Fig. 1d, e). Notably, Nlrc5 expression was abundant in the neointima rather than the media in the injured artery, and located more in the nucleus rather than

in the cytoplasm of α-SMA positive VSMCs. Despite a fraction of Nlrc5 co-localizing with endothelial cells marker CD31, there was no difference in endothelial Nlrc5 expression between sham and ligation groups (Supplementary Fig. 3). Using western blot, we further confirmed that Nlrc5 expression started to increase in carotid arteries at 1 week after the ligation and its expression was profoundly high at 2 weeks later, reaching approximately six-fold higher compared to the sham group, suggesting that Nlrc5 expression was induced in VSMCs during neointima formation (Fig. 1f, g). We next treated HASMCs with PDGF-BB (10 ng/ml), a potent stimulator of VSMCs and a key mediator in vascular injury, for different periods of time. NLRC5 expression started to slightly increase 1 h after stimulation and remained at a sustainable higher level after 12 h of stimulation (Fig. 1h). Since we observed that NLRC5 located in the nuclei of VSMCs in mouse carotid tissues in response to injury, we next examined NLRC5 expression in the cytoplasmic and nuclear fractions of PDGF-BB-stimulated HASMCs. NLRC5 increased at 6 h in the nucleus and was maintained up to 12 h following PDGF-BB stimulation (10 ng/ml). In contrast, the expression of NLRC5 in the cytoplasm of HASMCs was at a very low level and remained unchanged following PDGF-BB stimulation (10 ng/ml) (Fig. 1i). Although there was rare cytoplasmic localization, NLRC5 was predominantly expressed in the nuclei of HASMCs and increased in response to PDGF-BB treatment (10 ng/ml) (Fig. 1j).

**NLRC5 attenuated neointimal formation in vivo.** Given that NLRC5 was upregulated in VSMCs after carotid ligation, we hypothesized that NLRC5 contributed to neointimal formation after vascular injury. To test this hypothesis, we studied Nlrc5 whole body knockout mice (Nlrc5−/−). The strategy of knockout mouse generation is summarized in Fig. 2a and genotyping was performed in Nlrc5−/− and littermate Nlrc5+/+ mice. To investigate the effect of NLRC5 on neointimal formation, Nlrc5−/− and Nlrc5+/+ mice were subjected to vascular injury by carotid ligation for 3 weeks. We verified the success of Nlrc5 deletion in Nlrc5−/− mice and tested the specificity of Nlrc5 staining in ligated carotid arteries (Supplementary Fig. 4). NLRC5 deficiency significantly aggravated neointimal formation reflected by enlarged intima areas (Nlrc5+/+ $2.86 \pm 0.45 \times 10^4$ μm² vs. Nlrc5−/− $5.39 \pm 0.86 \times 10^4$ μm², $P < 0.01$ by Student's t-test) and increased intima/media ratios (Nlrc5+/+ $0.40 \pm 0.10$ vs. Nlrc5−/− $0.82 \pm 0.10$, $P < 0.01$ by Student's t-test, Fig. 2b–d). Furthermore, NLRC5 deficiency dramatically enhanced vascular hyperplasia reflected by the increased percentage of PCNA-positive nuclei, a marker of proliferation (Nlrc5+/+ $20.27 \pm 1.83\%$ vs. Nlrc5−/− $37.42 \pm 2.95\%$, $P < 0.01$ by Student's t-test, Fig. 2e, f).

We also measured systolic blood pressure (BP), heart rate, and analyzed the lipid profile to exclude potential confounding factors associated with vascular remodeling. Tail-cuff BP measurement showed no difference in systolic BP (Nlrc5+/+ $113.68 \pm 9.45$ mmHg vs. Nlrc5−/− $116.09 \pm 11.46$ mmHg) and heart rate between Nlrc5−/− and Nlrc5+/+ mice after 3-week complete carotid ligation (Supplementary Fig. 5A and B). Plasma total cholesterol, triglyceride, low density lipoprotein cholesterol, high density lipoprotein cholesterol, and fasting glucose levels were similar between two groups (Supplementary Fig. 5C and D).

In the hematopoietic system, NLRC5 is highly expressed in lymphocytes and myeloid cells, and NLRC5 deletion resulted in decreased CD8+ T cells number[24,25]. Based on these findings, we further analyzed T cells and myeloid cells after carotid ligation in Nlrc5−/− mice. Consistent with previous reports[24,26], flow cytometric analysis showed a significant reduction in the percentage of CD8+ T cells in splenocytes (Nlrc5+/+ $10.7 \pm 1.2\%$ vs. Nlrc5−/− $5.7 \pm 0.6\%$, $P = 0.010$ by Student's t-test,

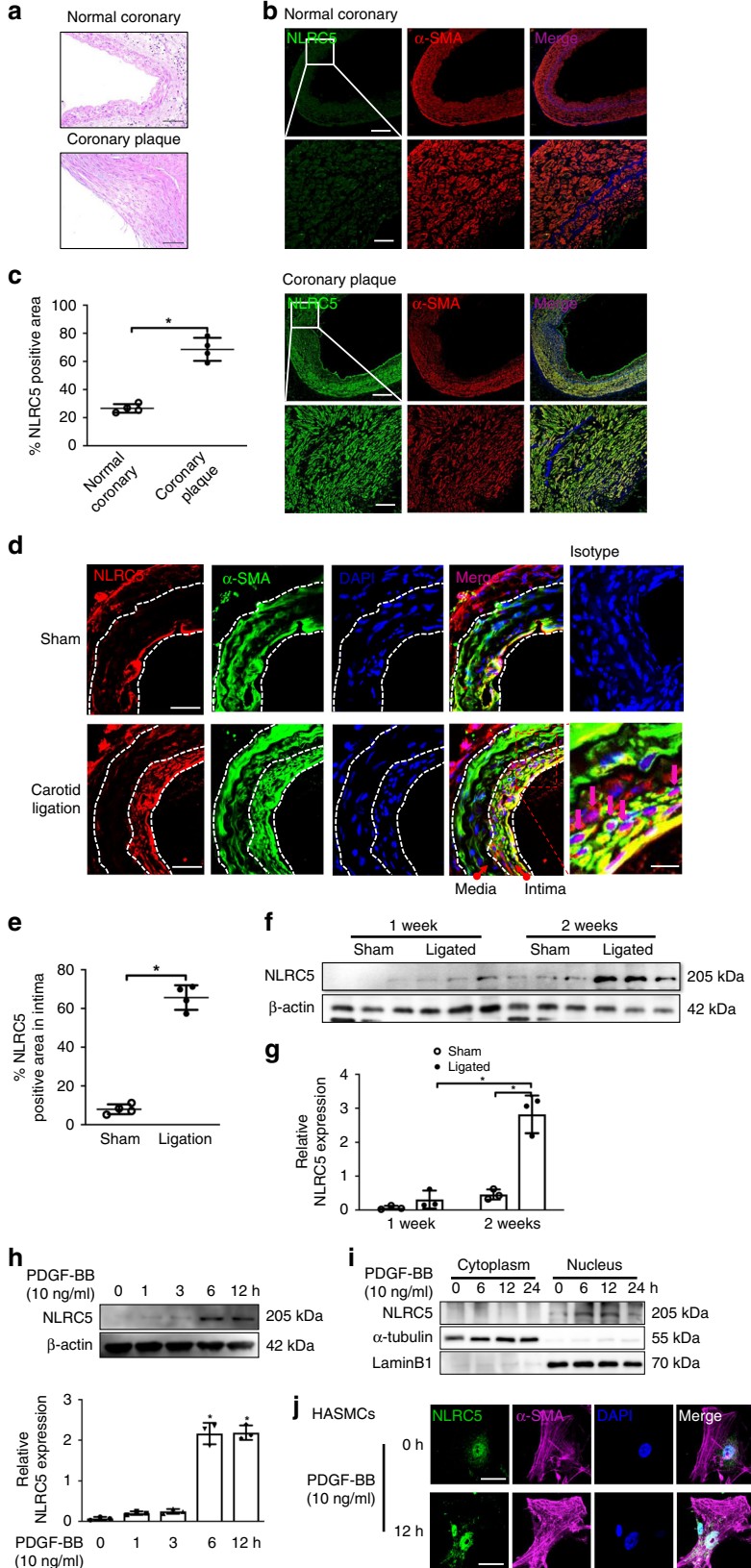

Supplementary Fig. 6A), and in peripheral blood ($Nlrc5^{+/+}$ 8.8 ± 0.3% vs. $Nlrc5^{-/-}$ 7.2 ± 0.3%, $P = 0.011$ by Student's $t$-test, Supplementary Fig. 6B) of $Nlrc5^{-/-}$ mice. No differences in $CD45^+CD11b^+Gr1^+$ myeloid cells were found in bone marrow, spleen, and peripheral blood between $Nlrc5^{-/-}$ and $Nlrc5^{+/+}$ mice (Supplementary Fig. 7A, B). Considering the fact that $CD45^+$ leukocytes infiltrated carotid arteries and modulated neointimal formation after vascular injury[27], we assessed the presence of $CD45^+$ cell population in ligated carotid arteries. There was no difference in the number of $CD45^+$ leukocytes in

**Fig. 1** NLRC5 is upregulated in human coronary plaque and in mouse ligated carotids. **a** Representative images of hematoxylin/eosin-stained normal coronary artery and coronary plaque. Scale bar: 100 μm. **b** Immunofluorescence staining shows that NLRC5 (green) is upregulated in vascular smooth muscle cells (VSMCs, stained in red) residing in coronary plaque. Scale bar: 100 μm (upper) and 20 μm (lower). **c** Quantitative analysis of the percentages of NLRC5-positive stained VSMCs in coronary arteries ($n = 3$ per group). Five fields per section from each sample are analyzed. **d** Immunofluorescence staining shows that NLRC5 (red) is constitutively colocalized with VSMCs (labeled with α-SMA in green) in the neointima layer following carotid ligation. Normal IgG isotype serves as negative control. Scale bar: 50 and 20 μm. **e** Quantitative analysis of the percentages of NLRC5-positive stained VSMCs in sham and ligated carotid artery from C57BL/6 mice after 3-week carotid ligation ($n = 5$ per group). Five fields per section from each sample are analyzed. **f, g** Western blots show the protein levels of NLRC5 in ligated carotids compared with sham carotids of C57BL/6 mice. The western blots are repeated in three samples after 1-week and 2-week carotid ligation. **h** NLRC5 is increased in human aortic smooth muscle cells (HASMCs) at indicated time points under PDGF-BB (10 ng/ml) stimulation. The western blots are repeated for three times. **i** NLRC5 is particularly expressed in the nuclei of HASMCs and upregulated in response to PDGF-BB (10 ng/ml) stimulation. **j** Representative immunofluorescence staining depicts that NLRC5 (green) is predominantly located in the nuclei (blue) of HASMCs with and without PDGF-BB stimulation. The cytoplasm of HASMC is stained with α-SMA (magenta). Scale bar: 10 μm. Data are presented as mean ± SD. Two-tailed Student's t-test is used to compare two groups (**c**, **e**, and **g**), and analysis of variance (ANOVA) followed by Bonferroni post hoc analysis is used to compare three or more groups (**h**). *$P < 0.05$. Original magnification, ×100 (**a** and **b**), ×400 (**b** and **d**) and ×630 (**j**). Source data are provided as a Source Data file

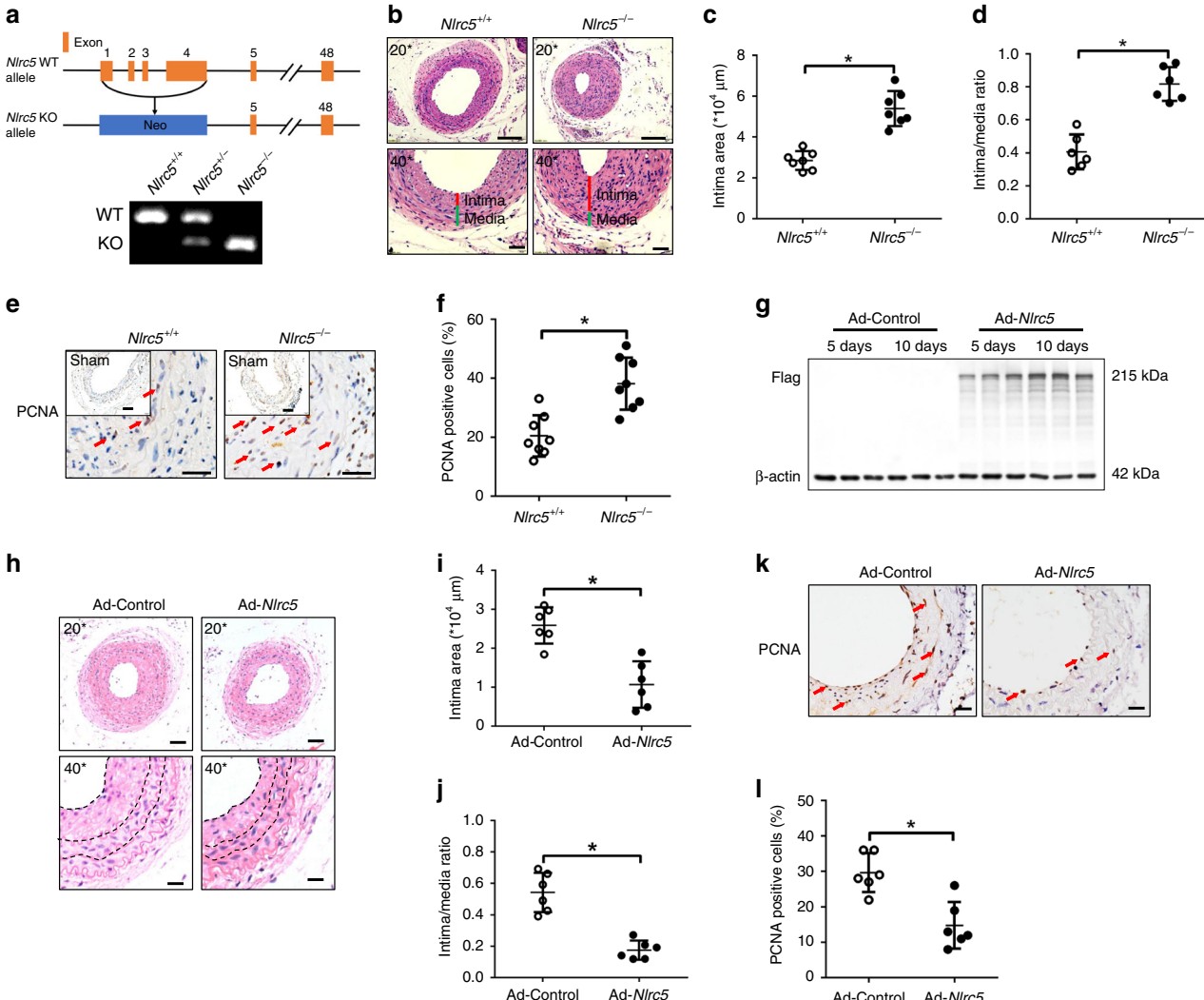

**Fig. 2** NLRC5 attenuates neointimal formation in vivo. **a** Schematic diagram of generating Nlrc5 knockout construct. Recombination of the Nlrc5−/− allele is examined by PCR in the Nlrc5+/+, Nlrc5+/−, and Nlrc5−/− mice. **b** Representative images of hematoxylin/eosin-stained Nlrc5−/− and Nlrc5+/+ mice carotid arteries at 3 weeks after carotid ligation. Scale bar: 100 μm (upper) and 50 μm (lower). **c, d** Quantification of the intima area and intima/media ratio in the histological sections ($n = 7$ per group). **e, f** Immunohistochemistry staining (red arrow) and quantitative analysis of PCNA-positive cells in the carotids ($n = 8$ per group). Five fields per section from each sample are analyzed. Scale bar: 100 and 20 μm. **g** Western blotting of Flag tag and β-actin in the ligated Ad-Control- and Ad-Nlrc5-transduced carotids at 5–10 days following carotid ligation. **h** Representative images of hematoxylin/eosin-stained carotid arteries transduced with Ad-Control or Ad-Nlrc5 at 3 weeks after carotid ligation. Scale bar: 50 μm (upper) and 20 μm (lower). **i, j** Quantification of the intima area and intima/media ratio in the histological sections ($n = 6$ per group). **k, l** Immunohistochemistry staining (red arrow) and quantitative analysis of PCNA-positive cells in the carotids transduced with Ad-Control or Ad-Nlrc5 ($n = 6$ per group). Five fields per section from each sample are analyzed. Scale bar: 20 μm. Data are presented as mean ± SD. Two-tailed Student's t-test is used to compare two groups (**c**, **d**, **f**, **i**, **j**, and **l**). *$P < 0.05$. Original magnification, ×200 (**b** and **h**) and ×400 (**b**, **e**, **h**, and **k**). Source data are provided as a Source Data file

ligated carotid arteries between $Nlrc5^{-/-}$ and $Nlrc5^{+/+}$ mice, implicating that depletion of NLRC5 did not affect leukocyte recruitment after carotid ligation (Supplementary Fig. 8).

In an opposite approach, we evaluated whether Nlrc5 overexpression suppresses neointimal thickening. Nlrc5 overexpression was induced by local transduction of Nlrc5 using adenoviruses. Overexpression of Nlrc5 was verified by both western blot analyses and immunofluorescence staining (Fig. 2g and Supplementary Fig. 9). Ad-Nlrc5 mice exhibited decreased intima areas (Ad-Control $2.59 \pm 0.47 \times 10^4 \mu m^2$ vs. Ad-Nlrc5 $1.07 \pm 0.60 \times 10^4 \mu m^2$, $P < 0.01$ by Student's $t$-test) and intima/media ratios (Ad-Control $0.54 \pm 0.12$ vs. Ad-Nlrc5 $0.18 \pm 0.06$, $P < 0.01$ by Student's $t$-test) 28 days after carotid ligation (Fig. 2h–j). The proliferation marker PCNA was remarkably decreased in carotid arteries after ligation with Nlrc5-overexpression (Ad-Control $38.24 \pm 2.83\%$ vs. Ad-Nlrc5 $21.06 \pm 3.96\%$, $P < 0.01$ by Student's $t$-test, Fig. 2k, l).

**NLRC5 prevented VSMC dysfunction in vitro.** VSMCs dedifferentiation, together with proliferation and migration, is a critical process for neointimal formation[28]. The MTS and Edu incorporation assays demonstrated that NLRC5 knockdown promoted HASMCs proliferation upon PDGF-BB stimulation (10 ng/ml) (Fig. 3a, b). MTS assay was performed on Ad-Control-transduced and Ad-NLRC5-transduced HASMCs. We found that NLRC5 overexpression prominently alleviated PDGF-BB-induced HASMCs proliferation (Supplementary Fig. 10A). To examine the effects of NLRC5 deficiency on VSMC migration, we performed scratch assay in HASMCs by silencing NLRC5. Migration of HASMCs was markedly enhanced after NLRC5 knockdown with 12-h PDGF stimulation (Fig. 3c, d), while much fewer Ad-NLRC5-overexpressing HASMCs migrated through scratch (Supplementary Fig. 10B and C).

Accumulating studies highlight that contractile VSMCs are capable of dedifferentiating into synthetic VSMCs in response to vascular injury and several extracellular stimuli[29]. VSMC phenotype switching coordinates with VSMC proliferation and migration. Myosin, α-SMA, and Calponin are considered as VSMC differentiation markers[30]. Consistent with the in vivo findings, NLRC5 knockdown reduced the expression of α-SMA, Calponin, and Myosin concomitant with increased expression of proliferative markers PCNA and Cyclin D1. These data indicated that NLRC5 depletion promoted VSMC dedifferentiation, a process of switching from a contractile to proliferative phenotype (Fig. 3e, f). In contrast, NLRC5 overexpression led to increased expression of α-SMA, Calponin, and Myosin, but decreased expression of PCNA and Cyclin D1 in HASMCs upon PDGF-BB stimulation (10 ng/ml) (Supplementary Fig. 10D and E).

Since VSMCs apoptosis was involved in neointimal formation, we also examined the influence of NLRC5 on apoptosis in HASMCs. No significant difference in early or late apoptosis was observed between scramble and NLRC5 siRNA groups (Supplementary Fig. 11A and B). In vivo TUNEL experiments also displayed similar apoptosis ratios in ligated carotid arteries between $Nlrc5^{-/-}$ and $Nlrc5^{+/+}$ mice (Supplementary Fig. 11C and 11D). In summary, these data implied that NLRC5 suppressed excessive VSMC proliferation, migration, and dedifferentiation upon PDGF-BB stimulation.

**Interaction between NLRC5 and PPARγ.** Current literature suggests a controversial role of NLRC5 in regulating inflammation via NF-κB-signaling pathway[31]. Interestingly, NLRC5 knockdown in HASMCs failed to regulate the phosphorylation of IκBα under the stimulation of PDGF-BB (10 ng/ml) (Supplementary Fig. 12A), which was consistent with the results that the

NLRC5 was rarely expressed in the cytoplasm (Fig. 1i). This led us to explore other possible signaling pathways downstream of NLRC5. It is reported that HLA is ubiquitously expressed in human tissues and cells, and NLRC5 acted as a transcriptional coactivator of MHC-I/HLA through recruitment of enhanceosome component RFX5 at SXY module[32]. Thus, we checked the expression of RFX5 and the concomitant changes in classical and unclassical HLA expression. RFX5 expression remained unchanged upon PDGF-BB stimulation (10 ng/ml) on the mRNA level and it was significantly lower in HASMCs than that in the human monocytic cell line THP-1, which served as a positive control (Supplementary Fig. 12B). It should be noted that, unlike MHC-II that is found only on antigen-presenting cells, MHC-I/HLA is constitutively expressed on the surface of all nucleated cells in mammalians. Compared with control group, neither classical HLA (HLA-A and HLA-B) nor unclassical HLA (HLA-E) expression level changed in HASMCs transfected with NLRC5 siRNA (Supplementary Fig. 12C–F). Based on the human inflammatory cytokine array, we found that NLRC5 silencing did not cause substantial alterations in the expression of MIF and Serpin E1 under the induction of PDGF-BB (Supplementary Fig. 12G–I). In contrast to the protective effect of NLRC5 in vascular remodeling, our prior study found that NLRC5 played a contradictory role in diabetic mice that loss of NLRC5 ameliorated DN[22]. Therefore, we hypothesized that different environments or stimuli may influence the function of NLRC5. To prove this, we stimulated VSMCs with high glucose or PDGF-BB and found that while PDGF-BB did not activate Smad2 phosphorylation, high glucose significantly stimulated Smad2 phosphorylation (Supplementary Fig. 13A–D). These effects were consistent with analogous findings in mesangial cells as described in our previous work[22]. Moreover, we determined the expression of Myosin, α-SMA, Cyclin D1, and PCNA expression in response to DN in different genetic mice. We found that the expression of proliferation and VSMC markers remained unchanged in kidneys of $Nlrc5^{+/+}$ diabetic mice compared with $Nlrc5^{-/-}$ diabetic mice (Supplementary Fig. 13E and F).

Given the structural homology of NLRC5 and the reported association between another NLR member class II transactivator (CIITA) and PPARγ[33], and also after excluding most of the traditional pathways that could potentially serve as downstream mediators of NLRC5, we then studied whether NLRC5 functioned through interaction with PPARγ. PDGF-BB stimulation (10 ng/ml) remarkably promoted the intrinsic interaction of NLRC5 with PPARγ in HASMCs within 6–12 h (Fig. 4a). Vice versa, immunoprecipitation with NLRC5 antibodies followed by immunoblot analyses showed an increase in the interaction of NLRC5 with PPARγ in the presence of PDGF-BB (Fig. 4a). NLRC5 and PPARγ co-localized in the nuclei of HASMCs by immunofluorescence staining (Fig. 4b). In parallel, we found that Nlrc5 expression co-localized with PPARγ in mouse carotid arteries after 1-week of carotid ligation through double immunofluorescent staining (Fig. 4c). These observations inspired us to that NLRC5 might directly bind to PPARγ. To address this question, myc-tagged NLRC5 and Flag-tagged PPARγ plasmids were co-expressed in HEK293T cells. Co-immunoprecipitation assays indicated a direct interaction between NLRC5 and PPARγ (Fig. 4d). The significant overlap of myc tag with Flag tag on confocal microscopy confirmed the colocalization of NLRC5 and PPARγ (Fig. 4e). When using a PPRE luciferase reporter system for testing PPRE activity, we further confirmed that NLRC5 deletion had a significant inhibitory effect on PPRE activity (Fig. 4f, g). On the other hand, transfection with NLRC5 overexpression plasmid induced PPRE luciferase activity in a dose-dependent manner (Fig. 4h). To confirm the effects of NLRC5-mediated regulation of the PPARγ

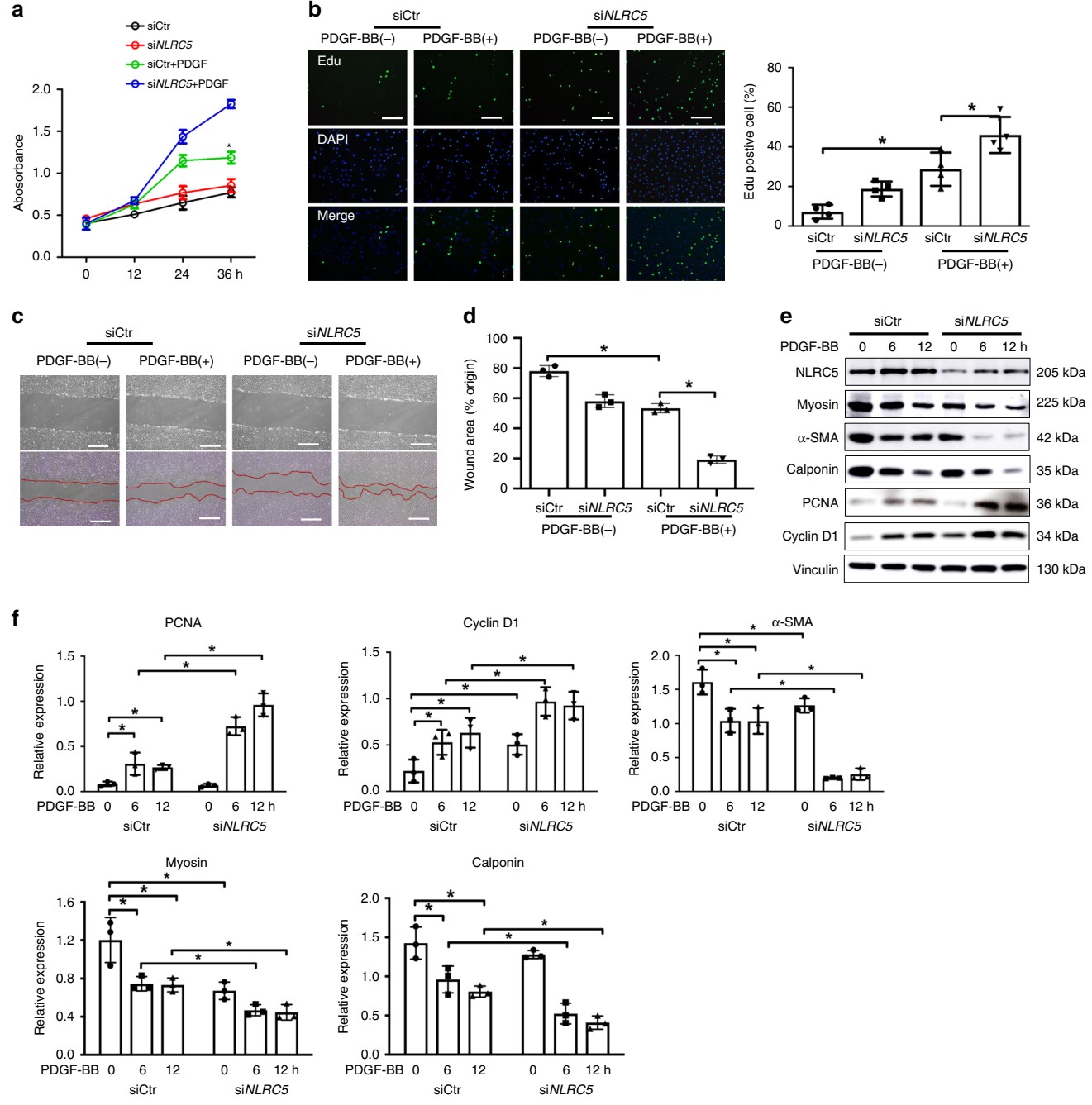

**Fig. 3** NLRC5 deficiency aggravates VSMC proliferation, migration, and dedifferentiation. **a** Human aortic smooth muscle cells (HASMCs) are transfected with scramble small interfering RNA (siCtr) or *NLRC5* small interfering RNA (si*NLRC5*) for 48 h. HASMC proliferation is measured by MTS assay in the presence or absence of PDGF-BB (10 ng/ml) at the indicated time points. *$P < 0.05$ vs si*NLRC5*+PDGF at 36 h after PDGF-BB stimulation. **b** Edu incorporation (green) is evaluated by fluorescence microscopy. Hoechst 33342 is used as a nuclear stain of HASMCs. Scale bar: 100 μm. **c**, **d** After transfection with siCtr or si*NLRC5* for 48 h, cell migration is assessed by scratch assay in HASMCs with or without PDGF-BB (10 ng/ml) stimulation for 12 h **c**. Scale bar: 100 μm. Wound area is analyzed by ImagePro Plus software **d**. **e**, **f** Representative western blotting of NLRC5, PCNA, Cyclin D1, α-SMA, Calponin, Myosin, and Vinculin in HASMCs transfected with siCtr or si*NLRC5* in the presence of PDGF-BB for 0, 6 and 12 h (10 ng/ml). Two-tailed Student's *t*-test is used to compare two groups **d**, and analysis of variance (ANOVA) followed by Bonferroni post hoc analysis is used to compare three or more groups **a**, **f**. Data are presented as mean ± SD from three independent experiments. *$P < 0.05$. Original magnification, ×100 (**b** and **c**). Source data are provided as a Source Data file

transcriptional network, we examined the expression of well-known PPARγ target genes related to proliferation and migration, including *CD36*, *AP2*, and *CITED2*[34–36]. In response to siRNA-mediated NLRC5 knockdown, the mRNA expressions of *CD36*, *AP2*, and *CITED2* were significantly decreased (Fig. 4i). Given that PPARγ/retinoid X receptors α (RXRα) heterodimers bind to

PPRE in the regulatory regions of target genes and activate gene transcription, we performed IP to determine whether the presence of NLRC5 regulated PPARγ/RXRα complex formation. However, absence of NLRC5 did not alter the PPARγ–RXRα interaction in HASMCs (Fig. 4j). To dissect whether NLRC5 serves as a PPARγ ligand without interfering with the

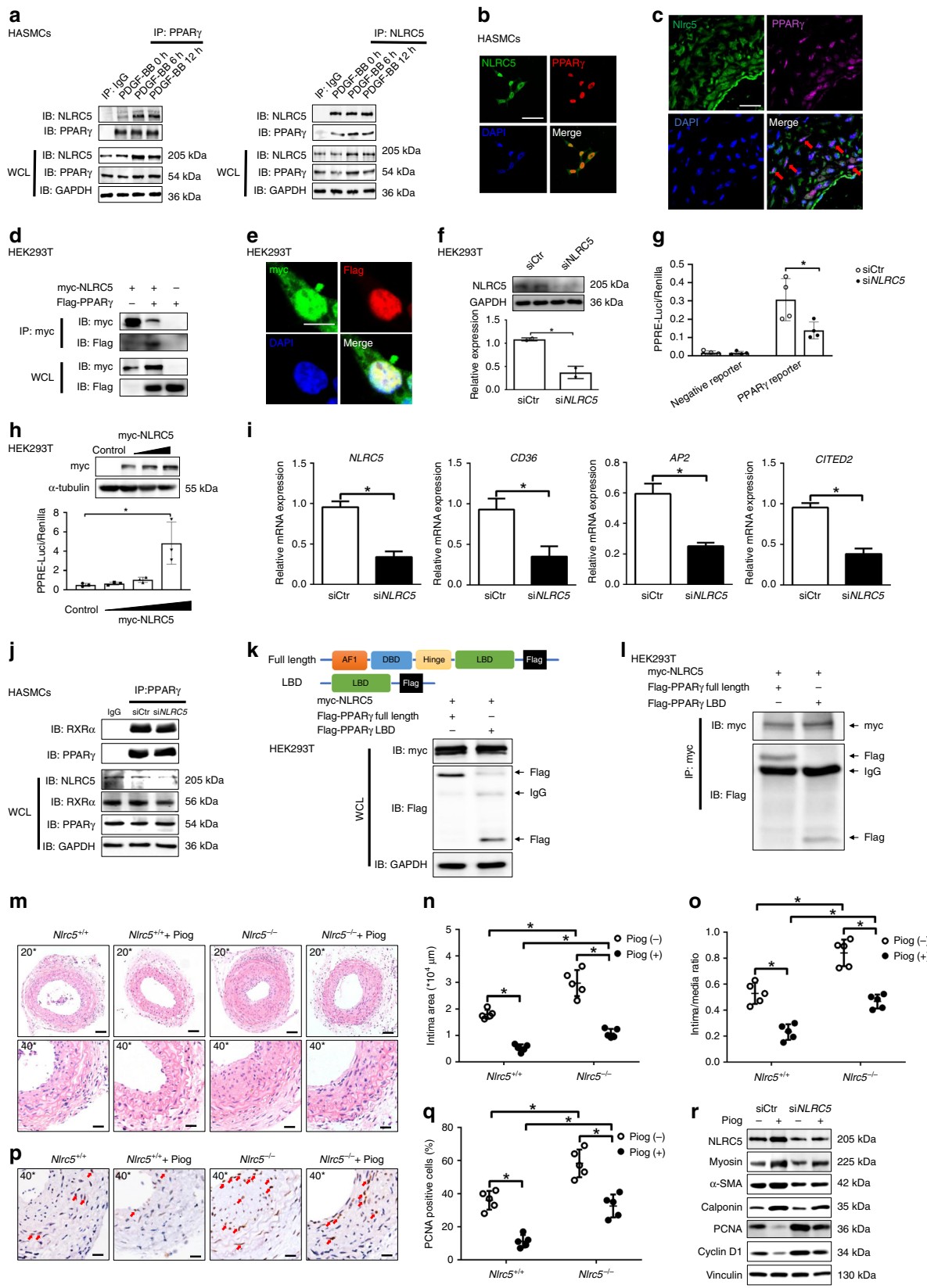

PPARγ/RXRα complex, we generated a Flag-PPARγ LBD and determined the intrinsic interaction between NLRC5 and LBD of PPARγ (Fig. 4k)[37]. Co-immunoprecipitation revealed the interaction of NLRC5 with the LBD of PPARγ, suggesting that the LBD domain, which was responsible for the binding of PPARγ

ligand, was required for the interaction with NLRC5 (Fig. 4l). These experiments indicate that NLRC5 predominantly interacts with PPARγ, acts as a PPARγ ligand and regulates PPRE-dependent gene transcription. Indeed, because of the residual levels of NLRC5 protein after knockdown of NLRC5, we could

**Fig. 4** NLRC5 directly interacts PPARγ and promotes PPARγ activity. **a** Co-immunopricipition of NLRC5 and PPARγ in HASMCs. **b** Immunofluorescence staining of NLRC5 (green), PPARγ (red), and nuclei (blue) in HASMCs. Scale bar: 20 μm. **c** Immunofluorescence staining of Nlrc5 (green), PPARγ (magenta), and nuclei (blue) in ligated cartids ($n = 3$ per group). Scale bar: 20 μm. **d** Co-immunoprecipition of myc and Flag in HEK293T co-transfected with myc-tagged NLRC5 and Flag-tagged PPARγ constructs. **e** Immunofluorescence staining of myc-tagged NLRC5 (green) and Flag-tagged PPARγ (red) in HEK293T cells. Scale bar: 10 μm. **f** Western blotting of NLRC5 and GAPDH in HEK293T cells transfected with siCtr or siNLRC5. **g** The activity of PPARγ response element (PPRE) is measured by luciferase reporter system. HEK293T cells are co-transfected with siCtr or siNLRC5 and PPARγ Cignal Reporter for 24 h. **h** The activity of PPRE in HEK293T cells co-transfected with empty or myc-tagged NLRC5 constructs and PPARγ Cignal Reporter. **i** Quantitative RT-PCR analyses of NLRC5, CD36, AP2, and CITED2 in HASMCs treated with PDGF-BB ($n = 3$ per group). **j** Co-immunopricipition of RXRα and PPARγ in HASMCs transfected with siCtr or siNLRC5. **k** Schematic diagram of full length PPARγ plasmid and the construct carrying ligand-binding domain (LBD) of PPARγ. Western blotting of HEK293T co-transfected with myc-tagged NLRC5 and Flag-tagged PPARγ, or Flag-taged PPARγ- LBD. **l** Co-immunopricipition of myc and Flag in HEK293T cells co-transfected with myc-tagged NLRC5 and Flag-tagged PPARγ, or Flag-taged PPARγ-LBD. **m–o** Representative images and quantification analyses of hematoxylin/eosin-stained $Nlrc5^{-/-}$ and $Nlrc5^{+/+}$ mice carotids at 3 weeks after carotid ligation with and without pioglitazone (10 nM) treatment ($n = 5$ per group). Scale bar: 50 μm (upper) and 20 μm (lower). **p, q** Immunohistochemistry staining (red arrow) and quantitative analysis of PCNA-positive cells in the carotids ($n = 5$ per group). Five fields per section from each sample are analyzed. Scale bar: 20 μm. **r** Representative western blotting of NLRC5, PCNA, Cyclin D1, α-SMA, Calponin, Myosin, and Vinculin in HASMCs. Original magnification, ×200 (**m**) and ×400 (**c**, **m**, and **p**). Two-tailed Student's t-test was used to compare two groups (**l**, **n**, **o**, and **q**). *$P < 0.05$. Source data are provided as a Source Data file

not entirely exclude the potential effect of NLRC5 on PPARγ/RXRα complex in HASMCs.

Activation of PPARγ is known to restrict vascular hyperplasia in response to vascular injury[38]. Simultaneously, our results depicting an interplay between NLRC5 and PPARγ inspired us to determine whether NLRC5 mitigated vascular hyperplasia through activation of PPARγ. We confirmed that the PPARγ agonist pioglitazone (10 nM) was able to moderately promote PPARγ expression in HASMCs and gradually enhanced PPRE activity as measured by luciferase assay in HEK293T cells (Supplementary Fig. 14A and B). Conversely, treatment of PPARγ antagonist T0070907 (100 nM) for 6 h inhibited PPARγ expression in HASMCs and repressed PPRE activity in HEK293T cells (Supplementary Fig. 14C and D). Second, we found that treatment of pioglitazone only partly rescued the excessive neointimal formation in $Nlrc5^{-/-}$ mice as compared to $Nlrc5^{-/-}$ mice without pioglitazone treatment (Fig. 4m–o). Concomitantly, pioglitazone significantly alleviated the in vivo proliferation as quantified by PCNA staining in ligated carotid arteries from $Nlrc5^{-/-}$ mice (Fig. 4p, q). To examine whether PPARγ contributed to NLRC5-mediated alleviation of VSMC proliferation and dedifferentiation in vitro, HASMCs were treated with PPARγ agonist pioglitazone (10 nM) with and without NLRC5 depletion. Pioglitazone treatment led to a significant reduction in PCNA and Cyclin D1 expression, accompanied with a recovery of α-SMA, Calponin, and Myosin expression, in the presence and absence of NLRC5 (Fig. 4r and Supplementary Fig. 15A). Conversely, the enhancement of α-SMA, Calponin, and Myosin expression in HASMCs transduced with Ad-NLRC5 was blocked in the presence of the T0070907, and accompanied by increases in PCNA and Cyclin D1 expression. These findings indicate that T0070907 counteracts the protective effect of NLRC5 on VSMC proliferation and dedifferentiation (Supplementary Fig. 15B and C). Collectively, the aforementioned in vivo and in vitro data suggest that NLRC5, at least in part, alleviates vascular remodeling through activation of PPARγ in VSMCs.

We next performed an in silico search within the promoter of NLRC5 gene for putative transcription factor-binding sites using binding profiles from JASPAR CORE database (jaspar.genereg.net). Interestingly, a predicted binding motif for PPARγ (JASPAR MA0066.1) was found in NLRC5 promoter with a score of 16.4 (Fig. 5a). Thus, we hypothesized that NLRC5 promoter sequences containing PPRE were a potential PPARγ-binding site (Fig. 5b). As an initial step, when HASMCs were treated with pioglitazone (10 nM), the mRNA and protein expression levels of NLRC5 were gradually increased in a time-dependent manner (Fig. 5c, d). Notably, the intrinsic interaction between NLRC5 and PPARγ existed without stimulation of the exogenous PPARγ ligand

(Fig. 5e). However, since the increased combination of NLRC5 and PPARγ paralleled the elevated expression of NLRC5 induced by pioglitzone, we could not conclude whether exogenous PPARγ ligand, pioglitazone, enhanced or competed with NLRC5–PPARγ interaction. Moreover, transient transfection with PPARγ siRNA in HASMCs and co-immunoprecipitation followed by immuno-blot analyses demonstrated that knockdown of PPARγ reduced NLRC5 expression and abrogated the interaction between NLRC5 and PPARγ as well (Fig. 5f). Furthermore, ChIP assay identified the PPARγ enrichment at the NLRC5 promoter in HASMCs under native conditions (Fig. 5g). We then used anti-PPARγ antibodies to immunoprecipitate protein/DNA complexes from HASMCs pretreated with pioglitazone for 0 and 12 h and quantitative PCR (ChIP-qPCR) amplified the putative PPRE of the promoter. The ChIP-qPCR studies showed that the PPARγ-binding to the NLRC5 promoter significantly increased in response to Pioglitazone (10 nM) (Fig. 5h), further supporting our hypothesis. We next generated a luciferase reporter construct carrying NLRC5 promoter (pGL3-NLRC5-promoter-Luci) and measured the luciferase activity. We found that PPARγ over-expression and pioglitazone facilitated NLRC5-luc activities in HEK293T cells (Fig. 5i, j). Based on the above results, we proposed that NLRC5 protected vascular remodeling through a positive feedback loop with PPARγ in VSMCs.

**The NACHT domain of NLRC5 interacted with PPARγ.** To better understand the underlying molecular mechanism, we generated different plasmids expressing different NLRC5 mutants (Fig. 6a) and evaluated their effects on PPARγ activity. NLRC5 consists of a tripartite structure, including an N-terminus, a central NACHT domain, and a C-terminal leucine-rich repeat domain (LRR)[39]. Unlike most of the NLR family members, the N-terminus of NLRC5 possesses an atypical CARD domain, also known as death-domain-like fold (DD). Based on the online structural data-base information (UniProtKB-Q86WI3, NLRC5_HUMAN, https://www.uniprot.org/uniprot/Q86WI3#structure), we defined DD domain of NLRC5 as amino acid (aa) 1–221 and the NACHT domain as aa 222–539. The mutants were individually co-transfected with the PPARγ vector in HEK293T cells, and cell lysates were then subjected to co-immunoprecipitation after 24-h transfection. We found that the DD mutant completely abolished NLRC5 and PPARγ interaction, whereas the NACHT domain of NLRC5 was essential to recognize and interact with PPARγ (Fig. 6b). To further validate the interaction of the NACHT domain with PPARγ, we applied confocal microscopy and detected that both myc-tagged DD + NACHT mutant and myc-tagged NACHT mutant were co-localized with Flag-tagged PPARγ in nuclei of

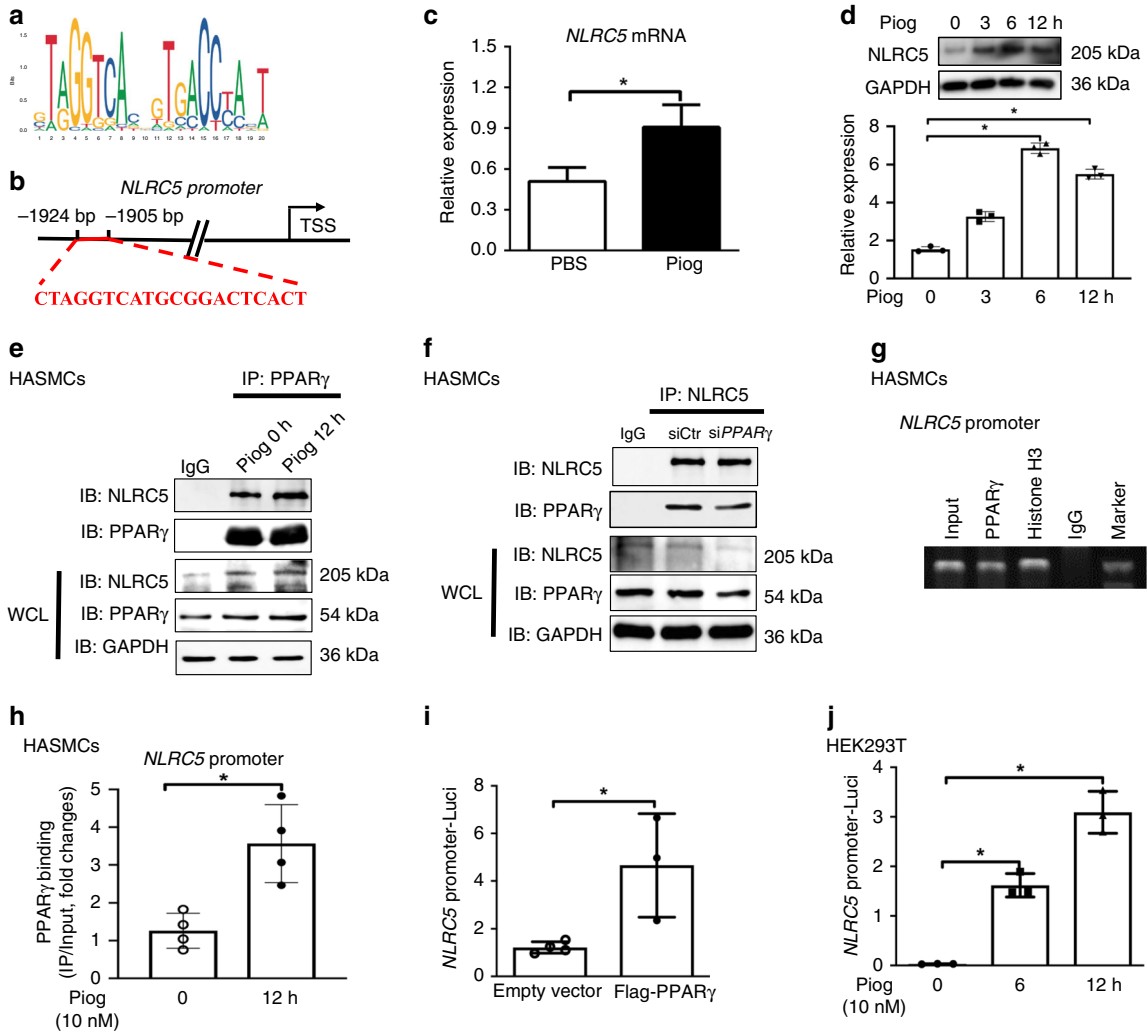

**Fig. 5** PPARγ binds to NLRC5 promoter and facilitate NLRC5 transcription. **a** The base sequence represents the consensus PPARγ-binding motif (JASPAR). **b** The sequence at *NLRC5* promoter indicates the putative PPARγ-binding element. **c** The mRNA expression levels of *NLRC5* in HASMCs in response to pioglitazone (10 nM) for 12 h are determined by quantitative RT-PCR. **d** Representative western blotting of NLRC5 and GAPDH in HASMCs treated with Pioglitazone (10 nM) for 0, 3, 6, and 12 h. **e** Co-immunopricipitation of NLRC5 and PPARγ in HASMCs with and without pioglitazone (10 nM) treatment. The lysates immunoprecipitated with anti-IgG serve as negative control. **f** Co-immunopricipitation of NLRC5 and PPARγ in HASMCs transfected with siCtr or si*PPARγ*. The lysates immunoprecipitated with anti-IgG serve as negative control. **g** Chromatin immunoprecipitation (ChIP) assay is performed to evaluate the binding of PPARγ to *NLRC5* promoter in HASMCs without PDGF-BB treatment. ChIP is performed with anti-PPARγ antibody, anti-histone H3, and normal rabbit IgG. The input represents electrophoresis of chromatin fragments without immunoprecipitation. **h** ChIP-qPCR is performed on HASMCs with antibodies to PPARγ and the target promoter region of *NLRC5* is amplified by qPCR. Data are presented as relative enrichment over input ± SD of three biological repeats. **i** HEK293T cells are co-transfected with empty or Flag-tagged PPARγ plasmid and pGL3-*NLRC5*-promoter-Luci plasmid for 24 h. Then the activity of *NLRC5* promoter is measured by Dual Luciferase Reporter system. **j** HEK293T cells are transfected with pGL3-*NLRC5*-promoter-Luci plasmid for 24 h and then treated with Pioglitazone (10 nM) for 6 and 12 h. Two-tailed Student's *t*-test is used to compare two groups (**c**, **h**, and **i**), and analysis of variance (ANOVA) followed by Bonferroni post hoc analysis was used to compare three or more groups (**d** and **j**). Data are presented as mean ± SD from three independent experiments. *$P < 0.05$. Source data are provided as a Source Data file

HEK293T cells (Fig. 6c). Consistent with these observations, NACHT, NACHT + DD, and ISO3 led to a remarkable increase in PPARγ activity (Fig. 6d) among the four established mutants. Moreover, compatible with increased protein expression, induction of NACHT, NACHT + DD, and ISO3 remarkably enhanced PPARγ activity and reached their peaks at 2 or 3 μg concentrations, while DD mutant did not affect PPARγ activity (Fig. 6e–h).

We then administrated adenovirus containing the NACHT domain of NLRC5 (Ad-NLRC5 NACHT) or empty adenovirus (Ad-Ctr) in HASMCs and assessed the effect of the NACHT domain on VSMC proliferation. The Edu incorporation assay showed that adenoviral overexpression of the NACHT domain substantially attenuated VSMC proliferation compared with

Ad-Ctr groups (Fig. 6i, j). Furthermore, after adenoviral overexpression of the NACHT domain of NLRC5 (Ad-NLRC5 NACHT), the expression of the PPARγ target genes *CD36*, *AP2*, and *CITED2* were markedly higher than those measured in Ad-Ctr-transduced HASMCs (Fig. 6k). To assess whether the suppressive role of the NACHT domain on VSMC proliferation in vitro similarly regulates neointimal formation in vivo, we ligated common carotid arteries in mice and performed local transduction of Ad-Control or Ad-Nlrc5 NACHT. Immunoblots for Flag tag were performed and the efficiency of localized delivery of adenovirus into carotid arteries was confirmed (Fig. 6l). In accordance with the in vitro observations, the cross-sectional areas of neointima (Ad-Control $1.29 \pm 0.38 \times 10^4$ μm²

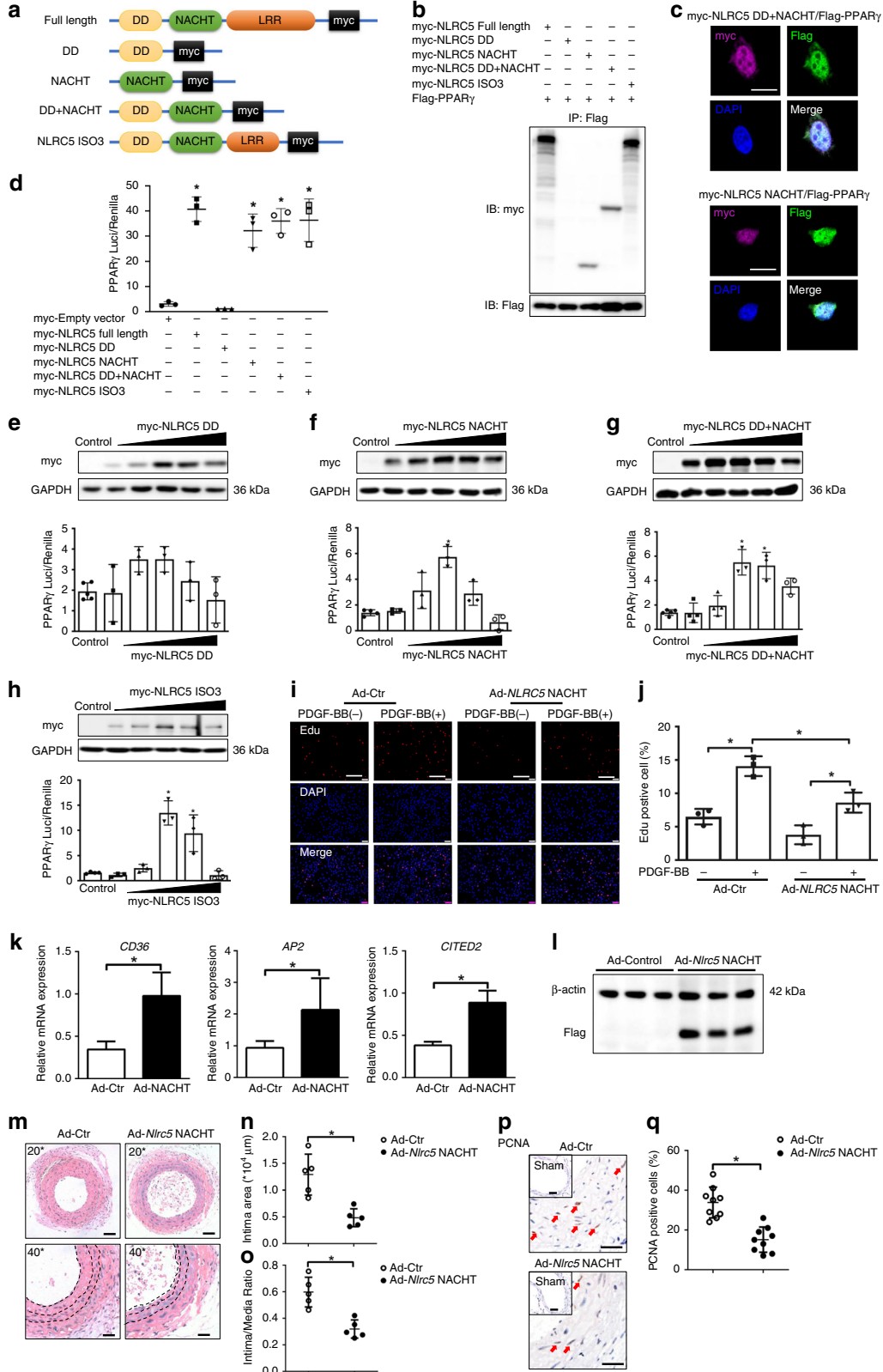

vs. Ad-*Nlrc5* NACHT $0.48 \pm 0.17 \times 10^4 \, \mu m^2$, $P < 0.01$ by Student's *t*-test) and intima/media ratios (Ad-Control $0.60 \pm 0.11$ vs. Ad-Nlrc5 NACHT $0.32 \pm 0.07$, $P < 0.01$ by Student's *t*-test) in ligated carotid arteries were markedly decreased in Ad-*Nlrc5* NACHT mice compared with Ad-Control mice (Fig. 6m–o). Immunohistochemistry analysis of PCNA-positive cells within

ligated carotid arteries indicated that VSMC proliferation of Ad-Control mice were more severe than that of Ad-Nlrc5 NACHT mice (Fig. 6p, q). Taken together, these results strongly suggested that the NACHT domain of NLRC5 was responsible for NLRC5 binding to PPARγ and positively regulating PPARγ activity.

**Fig. 6** Interaction and function analysis of NLRC5 domains. **a** Schematic diagram of four *NLRC5* mutant constructs. The structural domains are defined according to the structural database information (UniProtKB-Q86WI3, NLRC5_HUMAN, https://www.uniprot.org/uniprot/Q86WI3#structure).
**b** HEK293T cells are co-transfected with myc-tagged *NLRC5* or its mutants and Flag-tagged PPARγ construct. The interaction of *NLRC5* function domains with PPARγ is analyzed by co-immunoprecipitation. **c** Colocalization of myc-tagged *NLRC5* mutants (upper panel: DD + NACHT domain, lower panel: NACHT domain) and Flag-tagged PPARγ in HEK293T cells is visualized by confocal microscopy. Scale bar: 10 μm. **d** HEK293T cells are transfected with PPARγ Cignal Reporter, together with an empty vector, or full-length NLRC5, or its mutant constructs, and analyzed for luciferase activity.
**e–h** HEK293T cells are transfected with PPARγ Cignal Reporter, together with *NLRC5-DD* (**e**), *NLRC5-NACHT* (**f**), *NLRC5-DD + NACHT* (**g**), or *NLRC5-ISO3* (**h**) mutants at different concentrations (0, 0.5, 1, 2, 3 and 4 μg/ml), and analyzed for luciferase activity. **i, j** Edu incorporation (red) and nuclei (blue) in HASMCs transduced with Ad-Control or Ad-*Nlrc5* NACHT is evaluated by fluorescence microscopy. Scale bar: 100 μm. **k** Quantitative RT-PCR analyses of *CD36*, *AP2* and *CITED2* in HASMCs treated with PDGF-BB. **l** Western blotting of Flag and β-actin in the ligated Ad-Control- and Ad-*Nlrc5* NACHT transduced carotids at 3 weeks following carotid ligation. **m–o** Representative images and quantification analyses of hematoxylin/eosin-stained carotid arteries transduced with Ad-Control or Ad-*NLRC5* NACHT at 3 weeks following carotid ligation (*n* = 5 per group). Scale bar: 50 μm (upper) and 20 μm (lower). **p, q** Immunohistochemistry staining (red arrow) and quantitative analysis of PCNA-positive cells in the carotids transduced with Ad-Control or Ad-*Nlrc5* NACHT (*n* = 9 per group). Five fields per section from each sample are analyzed. Scale bar: 100 and 20 μm. Two-tailed Student's *t*-test is used to compare two groups (**j**, **k**, **n**, **o**, and **q**), and analysis of variance (ANOVA) followed by Bonferroni post hoc analysis is used to compare three or more groups (**e**, **f**, **g**, and **h**). Data are presented as mean ± SD from three independent experiments. *$P < 0.05$. Original magnification, ×100 (**i**), ×200 (**m**), ×400 (**m** and **p**) and ×630 (**c**). Source data are provided as a Source Data file

**Vascular NLRC5 was required for neointimal formation.** We further examined the possible contribution of vascular or hematopoietic *Nlrc5* in neointimal formation using mismatched bone marrow transplantation (BMT). *Nlrc5*$^{-/-}$ and *Nlrc5*$^{+/+}$ mice were sufficiently irradiated and reconstituted with either *Nlrc5*$^{+/+}$ or *Nlrc5*$^{-/-}$ BM (Fig. 7a). Firstly, BM cells from GFP-expressing mice were transplanted into *Nlrc5*$^{-/-}$ and *Nlrc5*$^{+/+}$ mice after irradiation. Using flow cytometry, the ratios of GFP-positive cells in BM, spleen, and peripheral blood from recipients exceeded 80%, which verified the success of BMT (Supplementary Fig. 16). In addition, we found few and comparable GFP-positive cells from BM infiltrating into neointima and media in carotids between *Nlrc5*$^{-/-}$ and *Nlrc5*$^{+/+}$ mice (Fig. 7b). Using genotyping and quantitative PCR in BM cells, splenocytes, and peripheral blood cells from *Nlrc5*$^{+/+}$ recipients, we further confirmed the high efficiency of BMT regardless of donor genotypes (Supplementary Fig. 17A–D). *Nlrc5*$^{-/-}$ background mice had equally increased intima areas and intima/media ratios when receiving *Nlrc5*$^{+/+}$ or *Nlrc5*$^{-/-}$ BM cells. Similarly, there was no difference in intima areas and intima/media ratios between *Nlrc5*$^{+/+}$ background mice receiving *Nlrc5*$^{+/+}$ or *Nlrc5*$^{-/-}$ BM cells (Fig. 7a–e). Accordingly, while depletion of *Nlrc5* in carotid parenchymal cells markedly aggravated VSMC proliferation indicated by PCNA-positive staining, depletion of *Nlrc5* in BM-derived cells did not efficiently affect vascular hyperplasia (Fig. 7f, g). Collectively, this mismatched BMT experiment excluded the contribution of hematopoietic *Nlrc5* in neointimal formation after carotid ligation, indicating an essential role for non-hematopoietic *Nlrc5* in vascular remodeling.

## Discussion

The findings in the present study bear therapeutic relevance based on: (1) accumulating data pointing to the essential role of immunity in the initiation and development of cardiovascular disease through vascular remodeling[40]; (2) the reported function of NLRC5 in the immune response[16]; and (3) our previous report of NLRC5 in facilitating DN, a disease where vascular remodeling is highly involved[22]. The current report extends these findings and establishes a direct role of NLRC5 in vascular remodeling. *Nlrc5*$^{-/-}$ mice exhibit more severe intimal hyperplasia as compared with *Nlrc5*$^{+/+}$ mice. Mechanistically, we identify that NLRC5 forms a positive feedback loop with PPARγ in VSMCs. Notably, NACTH domain is the essential domain of NLRC5 mediating PPARγ interaction. These findings demonstrating a key role of NLRC5 in orchestrating a complex process of vascular remodeling are summarized in Fig. 8.

Since the discovery of NOD-like family member NLRC5, it has been studied primarily in the immune system and under inflammatory conditions[16]. The innate immune system, such as TLR family members, have been involved in vascular injury and remodeling through regulating VSMC function[40]. Moreover, NLRC5 contributes to both innate immunity and inflammation[16]. Recent studies also revealed its role in chronic disease such as cancer[41]. Our recently published study showed that deficiency in NLRC5 ameliorated DN, a disease exemplified by inflammation, immune response, and vascular remodeling. In response to DN, NLRC5 regulated cellular effects by inhibiting high glucose-related NF-κB and TGF-β-signaling pathways[22]. We not only found enhanced NLRC5 expression in remodeled arteries, but also identified that NLRC5 was specifically located in the newly formed intima, co-localizing well with VSMC markers, rather than in adventitia or medial layers. Gain-of-function and loss-of-function studies revealed a distinct role for NLRC5 in vascular remodeling. Our in vitro data demonstrated that NLRC5 deficiency triggered VSMC dedifferentiation, proliferation, and migration, supporting the role of NLRC5 in regulating VSMC phenotypic switching. Because NLRC5 deficiency also affects the immune system, such as T cell subtype distribution as described by our group and others[24,25,42,43], we could not exclude the potential contribution of NLRC5 from immune cells on neointima formation. In addition, the neointima itself harbors macrophages and T cells, suggesting the potential role of innate and/or adaptive immunity in vascular remodeling[44]. Therefore, we generated chimeric mice with mismatched bone marrow to investigate the possible contribution of hematopoietic cells to NLRC5-related vascular remodeling. *Nlrc5*$^{-/-}$ recipient mice transplanted with *Nlrc5*$^{+/+}$ bone marrow exhibited equally increased intima areas and intima/media ratios compared with those that received *Nlrc5*$^{-/-}$ bone marrow. This suggests that the insufficiency of immune cells associated with the *Nlrc5* knockout mice has limited effect on intimal hyperplasia and that VSMCs are the main contributors to neointima formation. This is an unanticipated finding and it raises the possibility that the immune cell phenotype of *Nlrc5*$^{-/-}$ mice could possibly be regulated via the extra-hematopoietic system. The mismatched BMT experiment also highlighted that regardless of hematopoietic *Nlrc5* genotype, mice with the same genetic background, either *Nlrc5*$^{+/+}$ or *Nlrc5*$^{-/-}$, shared similar degree of vascular remodeling. It is worth pointing out that NLRC5 is also expressed in lung, liver, and gastrointestinal tract[15]. Therefore, despite the impressive enhanced expression of NLRC5 in VSMCs both in vivo and in vitro, the contribution of NLRC5 from other tissues could not be fully excluded.

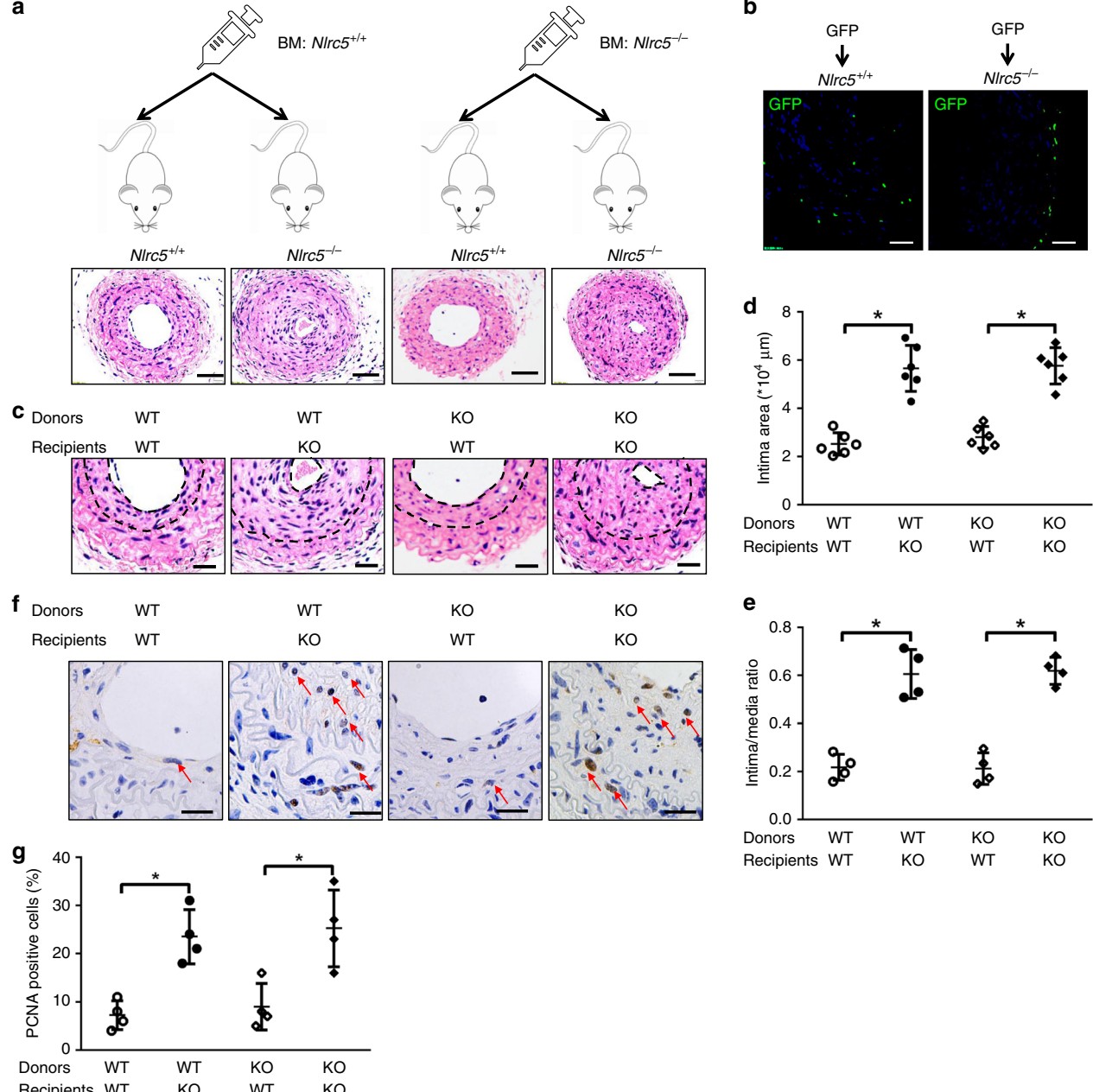

**Fig. 7** NLRC5 deficiency in BM-derived hematopoietic cells does not affect neointimal formation. **a** BM transplantation strategy. Bone marrow-derived cells from $Nlrc5^{+/+}$ and $Nlrc5^{+/+}$ mice were transplanted to $Nlrc5^{+/+}$ and $Nlrc5^{-/-}$ mice following irradiation. Scale bar: 100 μm. **b** Representative immunofluorescence images of ligated carotid arteries from $Nlrc5^{+/+}$ and $Nlrc5^{-/-}$ mice transplanted with GFP transgenic BM-derived cells. GFP signals (green) from BM-derived cells are detected in ligated carotids. The nuclei are stained with DAPI (blue) ($n = 3$ per group). Scale bar: 100 μm. **c** Representative images of hematoxylin/eosin-stained carotid arteries of $Nlrc5^{-/-}$ and $Nlrc5^{+/+}$ recipients harboring $Nlrc5^{-/-}$ or $Nlrc5^{+/+}$ BM-derived cells at 3 weeks after carotid ligation. Scale bar: 50 μm. **d, e** Quantification of the intima area and intima/media ratio in the histological sections ($n = 4$ per group). **f, g** Immunohistochemistry staining (red arrow) and quantitative analysis of PCNA-positive cells in the carotids of $Nlrc5^{-/-}$ and $Nlrc5^{+/+}$ recipients harboring $Nlrc5^{-/-}$ or $Nlrc5^{+/+}$ BM-derived cells at 3 weeks after carotid ligation ($n = 4$ per group). Five fields per section from each sample are analyzed. Scale bar: 20 μm. Two-tailed Student's $t$-test was used to compare two groups (**d**, **e**, and **g**). Data are presented as mean ± SD. *$P < 0.05$. Original magnification, ×200 (**a**) and ×400 (**b**, **c**, and **f**). Source data are provided as a Source Data file

The second significance of this study is that we characterize PPARγ as the downstream mediator of NLRC5 signaling in VSMCs. The cytoplasmic NLRC5 protein is highly involved in regulating NF-κB activities and modified by different deubiquitinases[45], while nuclear NLRC5 is thought to transactivate MHC I and related genes through its interaction with RFX5[32]. In our previous study in DN, we found that NLRC5 deficiency inhibited high glucose-related NF-κB activity in peritoneal macrophages and ameliorated inflammation. Interestingly, our current study shows that NLRC5 protein is highly expressed in the nucleus of stimulated VSMCs. This is consistent with the lack of influence of NLRC5 on NF-κB activation in stimulated VSMCs, since the latter is believed to function through NLRC5 in the cytoplasm. These data inform a shift in the mechanisms by which of NLRC5 in VSMCs functions—from a cytoplasmic-focused NF-κB pathway, towards an intra-nuclear factor pathway, such as

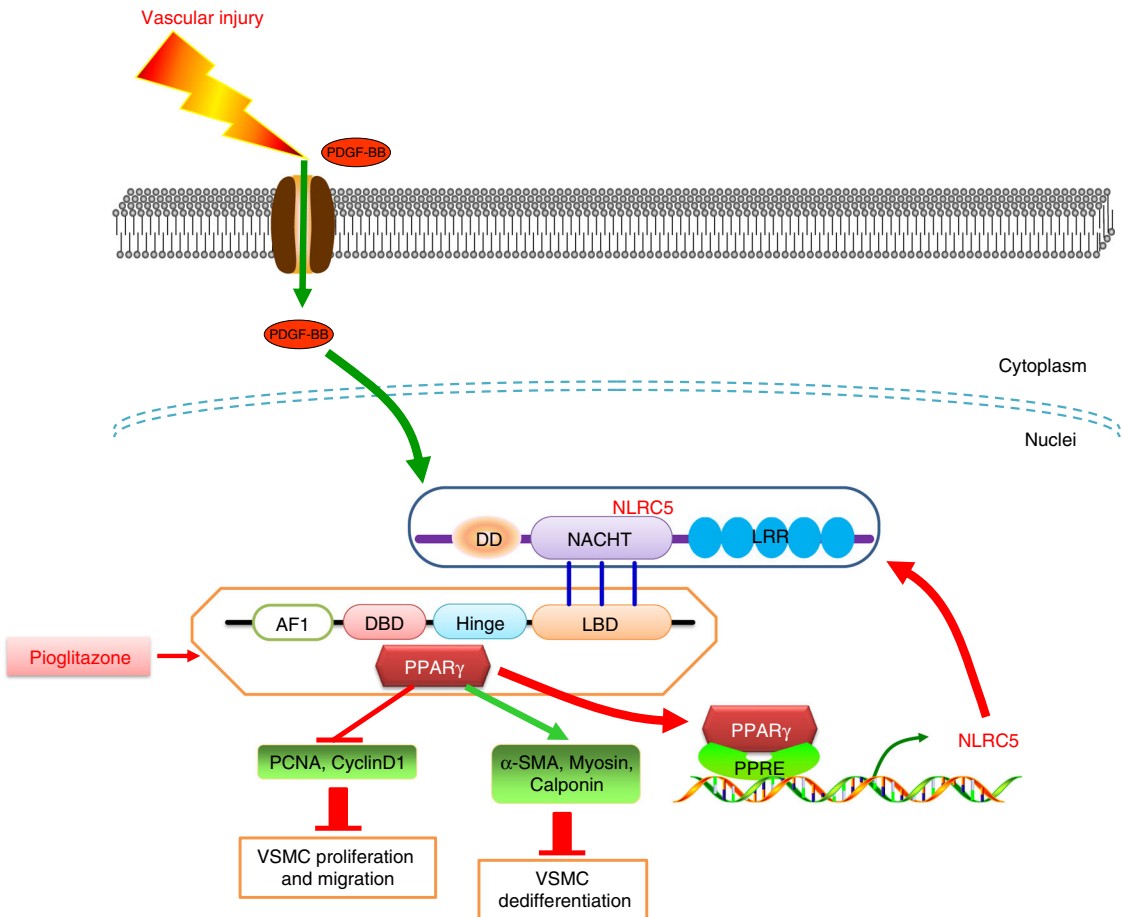

**Fig. 8** Proposed model of neointimal formation regulated by NLRC5-PPARγ feedback Vascular injury releases growth factor PDGF-BB that can activate NLRC5, which predominantly resides in the nuclei of VSMCs. NACHT domain of NLRC5 binds to ligand-binding domain (LBD) of PPARγ and facilitates PPARγ activity. PPARγ is in turn recruited at the promoter of NLRC5 and drives NLRC5 transcription. NLRC5-PPARγ positive feedback subsequently regulates downstream gene expression including PCNA, Cyclin D1, α-SMA, Myosin, and Calponin, and eventually suppresses VSMC proliferation, migration, and dedifferentiation and retards neointimal formation

transcription factor RFX5. However, RFX5 expression could not be induced in VSMCs upon PDGF-BB stimulation, thereby explaining in part the similar MHC I expression levels[32]. The above results imply that the effects of NLRC5 in VSMCs are beyond inflammation or immunity. It is known that PPARγ functions with the obligate heterodimer RXRα[46]. PPARγ/RXRα heterodimers binds to PPRE in the regulatory regions of target genes and activate gene transcription. Apart from this canonical PPARγ receptor, Zhang et al.[47] revealed that NF-κB p65 subunit and PPARγ formed an active transcription factor complex and cooperatively inhibited atherosclerotic progression. Nevertheless, our results show that absence of NLRC5 does not markedly disrupt PPARγ/RXRα heterodimer formation, but suppresses PPRE transcriptional activity. Furthermore, the observations that the LBD domain of PPARγ directly interacted with NLRC5, together with the in vivo and in vitro rescue experiments using PPARγ agonist pioglitazone, suggest that NLRC5 may serve as an endogenous ligand for PPARγ and provides a protective shield against vascular injury via activation of PPARγ.

Finally, our data substantiate that the NACHT domain is a key domain of the NLRC5 interaction with PPARγ, which potentially provides a promising therapeutic target for vascular remodeling-related diseases. NLRC5 is composed of DD domain, NACHT domain and LRR domain, and exhibits structural homology to CIITA[15,31]. Previous studies revealed that the DD domain

conferred transcriptional activity and regulated MHC-I/HLA transcription. The NACHT domain is known as the nucleotide-binding domain thought to be critical for nuclear import and transactivation activity[19], whereas the LRR domain is responsible for ligand binding. Besides finding the NACHT domain as an essential domain for the interaction between NLRC5 and PPARγ, we also identify that treatment with the PPARγ agonist pioglitazone significantly reduces VSMC proliferation and migration in HASMCs transfected with NLRC5 siRNA. Therefore, our study not only implicates NLRC5 as a therapeutic target for human disease characterized by vascular remodeling, but it also provides a precise domain that may serve as the foundation for future rational drug design. While PPARγ agonists and most of PPARγ target genes are proven protection in type 2 diabetes and cardiovascular diseases, a minority of PPARγ target genes, such as CD36, are reported to promote VSMC proliferation and tumor metastasis[48–52]. Given the side effect issues associated with PPARγ agonists, including tumor formation, bone fractures, weight gain, and fluid retention[53], targeting NLRC5 may provide a valuable means to modulate PPARγ signaling and potentially overcome these issues.

Collectively, this study demonstrates an essential role of NLRC5 in vascular intimal hyperplasia and establishes NLRC5 as a key transcriptional regulator in VSMCs uniquely through interaction with PPARγ.

## Methods

**Human artery sample collection**. Healthy coronary arteries were obtained from three patients undergoing trauma surgery without coronary plaques. Human coronary plaques were obtained from three patients undergoing coronary artery bypass grafting. The baseline characteristics of the patients are listed in Supplementary Table 1. The patient with Kawasaki disease was diagnosed according to the diagnostic criteria of the American Heart Association[23]. The study received approval by the Ethical Committee of Shanghai Tenth People's Hospital and experiments were conducted in compliance with all relevant ethical regulations. The written informed consent was collected from each patient and/or their relatives.

**Mice and complete carotid artery ligation**. Nlrc5 knockout (KO, Nlrc5−/−) mice (on C57BL/6 Background) were generated by Shanghai Biomodel Organism. Briefly, Nlrc5−/− mice (C57BL/6 background) were created by replacing exons 1–4 of the Nlrc5 gene with aneomycin-resistance gene[24]. The littermate wild type (WT, Nlrc5+/+) mice were used as control mice. Genotyping was conducted using the following primers: Nlrc5-forward 5′-CTGCCCAGGGAATTATGCTA-3′, Nlrc5-WT-reverse 5′-ATCCTGTGCTGCTCCTCAGT-3′ and Nlrc5-KO-reverse 5′-AATGTGTGCGAGGCCAGAG-3′. Ten-week-old male Nlrc5−/− and Nlrc5+/+ mice fed with normal chow diet were subjected to complete carotid ligation. In brief, mice were anesthetized with intraperitoneal injection of ketamine (80 mg/kg) and xylazine (5 mg/kg). The left common carotid artery was completely ligated with 6–0 silk suture just proximal to the carotid bifurcation. A similar procedure was performed but without ligation on the right common carotid artery serving as sham[54]. Peroxisome proliferator-activated receptor (PPARγ) agonist pioglitazone was suspended in water (20 mg/kg/day) and intraperitoneally injected in C57BL/6 mice 1 day-post carotid ligation[55]. The study received approval by the Animal Care and Use Committees of Shanghai Tenth People's Hospital for animal welfare. Experiments were conducted in compliance with the Guide for the Care and Use of Laboratory Animals published by the National Institutes of Health (NIH Publication, 8th Edition, 2011).

**Adenovirus production and localized delivery**. Adenoviruses for mouse Nlrc5 (Ad-Nlrc5), NACHT domain of Nlrc5 (Ad-Nlrc5 NACHT), and empty vector (Ad-Control) were purchased from GENECHEM Incorporation (China). For localized virus delivery, Ad-Nlrc5, Ad-Nlrc5 NACHT, or Ad-Control ($4 × 10^7$ pfu) were packaged by 70 μl Pluronic gel F-127 (Keygen, China) to extend virus contact time and delivery to bilateral carotid arteries at the time of ligation.

**Tail-cuff BP and heart rate measurement**. For systolic BP monitoring, mice were acclimated for at least 3 consecutive days before BP measurements and for one hour prior to performing the experiment. Then, all mice were encouraged to walk into the restraining tube, which was adjusted to prevent excessive movement. BP and heart rate measurement were carried out at the same time in a predetermined quiet area. BP measurements were taken three times from each mouse, and average number was reported in this study.

**Immunofluorescence staining**. Cells were seeded at $1 × 10^5$ cells/well on glass-bottomed culture dishes and then fixed with freshly prepared 4% paraformaldehyde for 15 min, followed by permeabilization with 0.2% Triton X-100 in PBS for 5 min. Human artery and mouse common carotid artery tissues were harvested and processed in optimal cutting temperature compound and sliced into 5 μm-thick sections. Paraformaldehyde-fixed sections, cryosections, and cells were incubated with anti-NLRC5 (ab105411, Abcam), anti-α-smooth muscle actin (α-SMA, ab7817, Abcam, 1:200), anti-PPARγ (sc-7196, Santa Cruz, 1:100), anti-CD31 (557355, BD Bioscience, USA, 1:100), anti-FLAG (ab49763, Santa Cruz, 1:100), and anti-myc (2276, Cell Signaling Technology, 1:100) overnight at 4 °C. Normal isotype IgG (sc2027, Santa Cruz) was used as negative control. After washing with phosphate-buffered saline (PBS), secondary antibodies (Alexa Fluor 647-conjugated goat anti-rabbit and Alexa Fluor 488-conjugated goat anti-mouse; Thermo Fisher Scientific, 1:200) were incubated for 1 h at 37 °C in the dark. Nuclei were labeled with DAPI (Vector Laboratories), and cells were visualized using an LSM710 laser confocal microscope (Carl Zeiss, Germany).

**Morphology and immunohistochemistry staining**. For morphological analysis, mice were perfused and fixed with 4% paraformaldehyde for 10 min. Paraformaldehyde-fixed carotid samples were embedded in paraffin blocks and 6 μm-thick sections were stained with hematoxylin and eosin (HE). For immunohistochemistry staining, sections were treated with microwave-based antigen retrieval using 10 mM sodium citrate buffer and then incubated with 0.3% hydrogen peroxide for 10 min to inactivate endogenous peroxide activity. After three washes in PBS, sections were incubated at 4 °C overnight with primary antibodies, anti-NLRC5 (ab105411, Abcam, 1:200), anti-PPARγ (sc-7196, Santa Cruz, 1:100) and anti-PCNA (sc-25280, Santa Cruz, 1:200), followed by incubation with corresponding biotin-conjugated secondary antibodies. Staining signal was detected using a standard ABC-peroxidase system (Vector Laboratories, USA). Subsequently, positive antibody binding was visualized using a DAB peroxidase substrate kit (Vector Laboratories, USA). Normal isotype IgG (sc2027, Santa Cruz, 1:400) was used as negative control. Images were captured by fluorescence

microscope (Olympus, Japan). Immunohistochemistry staining was quantified with Image-Pro Plus 6.0 software.

**Flow cytometry**. Peripheral blood cells from each mouse were obtained using heparin anticoagulant tubes before harvesting other tissues. Mice were then perfused with ice-cold PBS thoroughly before spleens, tibia, and femur bones were harvested. Bone marrow cells from tibia and femur bones were flushed out using cold Roswell Park Memorial Institute (RPMI)-filled syringe, whereas cells were isolated from spleens using mesh. The suspensions of bone marrow cells or splenocytes were obtained after going through a 0.45 μm strainer, and red blood cells were lysed in the dark (10 × FACS Lysing Solution, BD Pharmingen, USA). Cell suspensions were blocked with 1% BSA solution for 15 min at 4 °C and then stained with corresponding fluorescently labeled antibodies, diluted in 0.1% BSA solution at the indicated concentration (Supplementary Table 2).

**Cell culture and small interfering RNA (siRNA) transfection**. Human aortic smooth muscle cells (HASMCs) from ScienCell Research Laboratories were isolated from human aorta and cultured in smooth muscle cell medium (SMCM, Cat. #1101, ScienCell, USA) supplemented with 2% FBS, 1% SMCGS, and 1% penicillin and streptomycin. Experiments were performed using cells from passages 3–6. For reproducibility, we purchased and applied two batches of HASMCs in the following experiments. HEK293T cells were purchased from China Center for Type Culture Collection (Wuhan University, Hubei, China). Cells were cultured in DMEM supplemented with 10% fetal bovine serum, 100 U/ml penicillin, and 100 g/ml streptomycin and maintained at 37 °C in 5% $CO_2$. The duplex siRNA targeting NLRC5 (SASI_Hs02_00359503) and scramble siRNA (siCtr) were purchased from Sigma incorporation (Sigma, USA). The siRNA-targeting PPARγ was purchased from Sangon Biotech (Shanghai, China) and the sequences are as followed: sense- 5′-CUG GCC UCC UUG AUG AAU AUU-3′, antisense-5′-UAU UCA UCA AGG AGG CCA GTT-3′. HASMCs, seeded in a six-well plate at the density of $1.5 × 10^6$ cells/well, were transfected with 50 nM siRNA using 3 μl of RNAiMAX (Thermo Fisher, USA) in OPTI-MEM (Thermo Fisher, USA) for 24 h. Pioglitazone (CDS021593, Sigma), a therapeutic drug for diabetes, was used as an agonist of PPARγ. T0070907 (S2871, Selleck Chemicals, USA) was used as an antagonist of PPARγ.

**Cell proliferation analysis and scratch assay**. HASMC proliferation was assessed by 5-ethynyl-2′-deoxyuridine (Edu) incorporation assay (Catalog #C10337, Thermo, USA). HASMCs seeded in 24-well plates were washed with PBS and incubated with Edu-labeling mixture (10 mM) for 12 h accompanied with recombinant human platelet-derived growth factor (PDGF-BB, Catalog #220-BB, R&D Systems, America) stimulation. Cells then were fixed, permeabilized, and Edu incorporation was detected according to the manufacturer's instructions. Images were captured by fluorescence microscope (Olympus, Japan). Data were presented as ratio of Edu-positive cells to total cells. We also applied CellTiter 96 Aqueous One Solution (MTS, Promega, USA) to assess HASMC proliferation. HASMCs were incubated with 20 mM MTS solution for 2 h and measured at 490 nm absorbance by an automatic microplate reader (SpectraMaxi3, Molecular Devices, USA).

In vitro migratory activity of HASMCs was measured using a scratch assay. HASMCs were seeded into six-well plates at $1 × 10^5$ cells/well and cultured in SMCM supplemented with 2% FBS, 1% SMCGS, and 1% penicillin and streptomycin. When the cells reached 90% confluence, the growth medium was replaced by SMCM with 0.2% FBS. After 12-h starvation, the wound was made by scraping the cell monolayer with a 200 μl pipette tip across the center of the well.

**Measurement of cell apoptosis**. Measurement of apoptotic HASMCs was determined by flow cytometry-based Annexin V-FITC/PI staining (556547, BD Bioscience, USA). The double-negative cells (viable), Annexin V single-positive cells (early apoptosis) and double-positive cells (necrosis) were analyzed using FlowJo Software (V10.0.7, USA). Apoptotic cells in carotid arteries were assessed using TUNEL assay (11772465001, Roche, Germany). The number of TUNEL positive cells was counted in 10 randomly selected fields in each carotid sample under ×400magnification.

**Plasmid construction**. Mutant NLRC5 domain constructs and mutant PPARγ constructs were purchased from Shanghai Genechem Co., Ltd. All the following plasmids were generated from pGV219 vector with myc tag or pGV141 vector with Flag tag, including myc-NLRC5 Full length, myc-NLRC5 DD, myc-NLRC5 NACHT, myc-NLRC5 DD+NACHT, Flag-PPARγ, and Flag-PPARγ ligand-binding domain (LBD). Myc-NLRC5 ISO3 vector with deletion of partial LRR fragments was purchased from Addgene Incorporation and generated by Neerincx et al[56].

**Real-time quantitative RT-PCR**. Total RNA was isolated from tissues or cells with Trizol reagent (Thermo Fisher, USA). Purified RNA (500 ng) was reverse-transcribed using PrimerScript RT Reagent Kit (Takara, Japan) and quantitative RT-PCR was performed on 1 μg of cDNA product using FastStart Universal SYBR Green Master (Roche, USA) on a Roche Lightcycler. Information of primers is presented in Supplementary Table 3.

**Protein extraction and western blot**. Carotid artery tissue lysate or whole cells from in vitro experiments were prepared by 1 × cell lysis buffer (Cell Signaling Technologies, USA) containing protease inhibitors (Cat. 04693159001; Roche Molecular Biochemicals, USA). Given the low abundance of protein collected from mouse carotid artery, we mixed carotid arteries from two individuals under equal condition for the protein extraction. Lysates were cleared by centrifugation. Nuclear and cytoplasmic preparations were separated by NE-PER Nuclear and Cytoplasmic Extraction Reagents according to manufacturer's instructions (Cat. 78833; Thermo Fisher, USA). Briefly, HASMCs were detached using trypsin and washed with PBS. The cell lysis named CER I was added to cell extractions. Cytoplasmic fractions were collected via vortex, incubation with CER II and centrifugation. The insoluble pellets were then suspended in ice-cold NER. Nuclear fractions were collected via vortex and centrifugation without incubation with CER II. Protein concentrations were determined using bicinchoninic acid protein assay. Because the total protein amount in mice carotid tissues is very low, each lane comprised protein extracts of at least two carotids from mice in the same group when processing SDS–PAGE. Proteins were separated by SDS–PAGE, transferred to polyvinylidene fluoride (PVDF) membranes and incubated overnight at 4 ℃ with primary antibodies including anti-PPARγ (sc-7196, Santa Cruz, USA, 1:500), anti-RXRα (3085, Cell Signaling Technology, USA, 1:1000), anti-PCNA (ab29, Abcam, USA, 1:1000), anti-Cyclin D1 (2978, Cell Signaling Technology, USA, 1:1000), anti-α-SMA (ab5694, Abcam, USA, 1:1000), anti-Calponin (ab46794, Abcam, USA, 1:1000), anti-Myosin (ab53219, Abcam, USA, 1:2000), anti-NLRC5 (ab117624, Abcam, USA, 1:500), anti-GAPDH (60004-1-Ig, Proteintech, USA, 1:10,000), anti-Lamin B1 (66095-1-Ig, Proteintech, USA, 1:2000), anti-α-tubulin (ab52866, Abcam, USA, 1:10,000), anti-β-actin (60008-1-Ig, Proteintech, USA, 1:5000), anti-vinculin (sc73614, Santa Cruz, USA, 1:2000), anti-FLAG (ab1162, Abcam, USA, 1:2000), and anti-myc (2276, Cell Signaling Technology, USA, 1:2000). Primary antibodies were then incubated with secondary antibody for one hour and bends were visualized using chemiluminescence (ECL, TANON, China) and viewed under Amersham Imager 600 system (GE Healthcare, USA). Uncropped scans of the most important immunoblots are supplied in the Source Data file.

**Co-immunoprecipitation**. HASMCs or HEK293T cells were lysed in 1 × cell lysis buffer (Cell Signaling Technologies, USA) containing protease inhibitors (Cat. 04693159001; Roche Molecular Biochemicals, USA). After centrifugation, 500 μg of cell lysate was incubated with 5 μg of the indicated primary antibodies at 4 ℃ overnight. The lysate immunoprecipitated with anti-IgG serve as negative control. The immune complexes were then purified by 20 μl of protein A/G agarose (sc-2003, Santa Cruz, USA) at 4 ℃ for 3 h, centrifuged and washed by ice-cold cell lysis buffer. The immunoprecipitated protein was further analyzed by immunoblot.

**Dual-Luciferase assay**. HEK293T cells were maintained in DMEM supplemented with 10% FBS and 1% penicillin and streptomycinin, then seeded in 12-well plates were co-transfected with 1 μg of PPARγ response element (PPRE) Reporter mixture (PPRE Luciferase Reporter: Renilla construct = 40:1, Qiagen, Germany) and 1 μg of NLRC5 or mutant NLRC5 domain constructs using Lipofectamine 2000 (Thermo Fisher, USA). The amplified fragments of NLRC5 promoter (Gene accession NM_001330552, pGL-NLRC5 promoter) were inserted into pGL3-basic vector (Promega, USA) and were sequenced. Cellular lysates were collected 24 h after transfection using passive lysis buffer. PPRE and NLRC5 promoter Luciferase activity was measured using Dual Luciferase Reporter Assay System (Catalog #E1910, Promega, USA) by SpectraMaxi3 reader.

**Bone marrow transplantation**. Recipient mice were irradiated with 9 Gy of radiation at least 6 h prior to injection. On the day of BMT, donors were sacrificed and disinfected. Femur and tibia were collected in sterilized PBS on ice, and muscle tissue was thoroughly cleaned. Both ends of the bones were cut off and bone marrow cells were flushed out using 1 ml syringe needle filled with RPMI medium. Bone marrow was filtrated through 0.45 μm strainer. Cells were centrifuged and then suspended with 1 ml RPMI 1640. After being counted, cells were transplanted to recipients through tail vein injection. Each recipient was injected with $1 \times 10^7$ cells with an injection volume of 400 μl.

**Cytokine profiling array**. Thirty-six different inflammatory markers of HASMCs at the proteins level were examined using a Human Cytokine Array (R&D System ARY005B) according to the manufacturer's instructions. Briefly, membranes were incubated with 100 μg of total protein lysate and a cocktail of biotinylated antibodies overnight at 4 ℃. Following three washes, membranes were incubated in the presence of 2 ml (1:2000 dilution) of streptavidin–horseradish peroxidase (HRP) for 30 min at room temperature, and the presence of immunocomplexes was detected by staining with 3,3′-diaminobenzidine (DAB) chromogen. Arrays were scanned and pixel density was quantified using ImageJ software (V1.49, NIH).

**Chromatin immunoprecipitation (ChIP) assay**. ChIP assay was performed using a SimpleChIP Enzymatic Chromatin IP kit (cat No. 9003; Cell Signaling Technologies). HASMCs were treated with 20 ng/ml PDGF-BB for 12 h at 37 ℃ and then cross-linked with 37% formaldehyde at a final concentration of 1% at room temperature for 10 min. Fragmented chromatin was treated with nuclease and subjected to sonication.

ChIP was performed with rabbit anti-PPARγ antibody (ab45036, Abcam, 1:100), rabbit anti-histone H3 (a technical positive control; 1:50) (4620, Cell Signaling Technologies), and normal rabbit IgG (2729, Cell Signaling Technologies). After DNA purification, immunoprecipitated DNA was detected using standard PCR. Information of primers predesigned for ChIP is presented in Supplementary Table 3.

**Statistical analysis**. Data were presented as mean ± standard deviation (SD). A two-side, unpaired Student's t-test was used to analyze the difference between two groups of data with normally distributed variables. Mann–Whitney test was used in non-normally distributed variables. Differences across three or more groups were tested with one-way ANOVA followed by a post hoc analysis with Bonferroni test. A P-value ≤ 0.05 was defined as statistical significance.

**Reporting summary**. Further information on research design is available in the Nature Research Reporting Summary linked to this article.

## Data availability

The structural domains of NLRC5 are defined according to the structural database information (UniProtKB-Q86WI3, NLRC5_HUMAN, https://www.uniprot.org/uniprot/Q86WI3#structure). The base sequence representing the consensus PPARγ binding motif is acquired from JASPAR (jaspar.genereg.net). All the data supporting the findings of this study are available within the article and its Supplementary Information files and from the corresponding author upon reasonable request. The source data underlying all Figures and Supplementary Figures are provided as a Source Data file. A reporting summary for this article is available as a Supplementary Information file.

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

## Acknowledgements

We thank patients who participated in this study. This study is supported by Grant Nos. 81670230, 81670746, 81800424, and 81370391 from the Chinese National Natural Science Foundation, by Grant No. 1501219165 from Fundamental Research Funds for the Central Universities, and by National Institutes of Health grants (HL115141, HL134849, and GM115605). We are grateful to Dr. Xinghui Sun from the University of Nebraska for his critical discussions and comments on the manuscript.

## Author contributions

P.L. performed the animal experiments and flow cytometry, conducted co-immunoprecipitation assays, and immunoblots, analyzed data and interpreted the results. W.J. designed and performed experiments, analyzed data, and interpreted results. X.X. conducted the immunofluorescence staining of human artery samples, analyzed, and interpreted results. W.K. performed the histology of mice artery samples. Q.Y. performed immunoblots and quantitative PCR, and analyzed the data. H.H. and D.L. conducted bone marrow transplantations. W.W. helped to design and interpret the experiments. M.W.F. helped interpret the data and wrote the manuscript. J.Z. designed experiments, performed the animal experiments, conducted luciferase and ChIp assays, interpreted results and wrote the manuscript. Y.X. supported the initiation of the study and helped design the study. W.P. conceived the project, designed experiments, analyzed data, interpreted results, and wrote the manuscript.

## Additional information

**Competing interests:** The authors declare no competing interests.

