## [Peer Review File · Nature Communications]

Reviewers' comments:

Reviewer #1 (Remarks to the Author):

This is a very interesting paper describing a role for NLRC5 as an inhibitor of neointima formation through an interaction with PPARG. The data and implications are highly significant and novel. The paper would be improved by some additional data flushing out mechanisms of the NLRC5/PPARG pathway as defined in the comments below.

Major Concerns

1. Given the large, easily detectable increase in Nlrc5 by western blot (Figure 2G), it would be attractive to demonstrate whether this was expressed in endothelial cells, smooth muscle cells or adventitia after the gene transfer experiment. The use of the FLAG tag should easily facilitate this.
2. The efficacy of the siNLRC5 in Figure 4E is unclear. Can you quantify the knock down in each sample used to measure Luc activity?
3. Page 20- you say "The ChIP-qPCR results showed that PPAR γ -binding to NLRC5 promoter greatly increased in response to PDGF-BB stimulation (Figure 4L)". Figure 4L does not show the ChIP response to PDGF, it shows response to Pioglitazone.
4. You are suggesting a forward feedback mechanism whereby PPARG activation increases PPARG-mediated transcription of NLRC5 which then binds to PPARG further facilitating PPARG transcriptional activity. A) Is there evidence for the induction of bona fide PPARG target genes in HASMC that is mediated by NLRC5/PPARG complex? B) Does the complex require a PPARG ligand to form? C) Does NLRC5 participate in the PPARG/PPRE complex? D) Does NLRC5 act as a PPARG ligand?
5. The experiment in Figure 4O-P concluding that NLRC5 protected vascular remodeling through a positive feedback loop with PPAR γ in VSMCs would be much stronger if it was coupled to an experiment using an PPARG antagonist or siRNA targeting PPARG. This would show that the protective effects of NLRC5 are mediated by, or require PPARG.

6. The data in Figure 5 elegantly show that the NACHT domain is all that is required to bind to PPARG and mediate an induction in PPARG activity. It would be attractive to confirm this effect on a bona fide PPARG target gene, perhaps CD36 or AP2, the latter is ubiquitously expressed in many cell types and is quite abundant in SMC.

7. Like you showed in Figure 2H-L, is over-expression of the NACHT domain sufficient to mediate a decrease in intimal area and HASMC proliferation?

Minor Concerns

8. It would be preferred to perform BP measurements with an assay which is more reliable than tail cuff as there appears to be a trend toward an increase in the KO mice (Figure S3A). If showing tail cuff, only show the SBP as the DBP is a calculated variable not a measured variable.

9. Please check if the correct blots for p-IkB and IκB are used in Figure S9A as they look identical and there does not appear to be induction of P-IκB by PDGF.

10. Whenever possible, it would be more attractive to use increased transparency in the reporting of data by substituting bar/plunger graphs with dot/whisker plots.

11. Discussion: It would be worthwhile citing other studies showing the direct association between PPARG and other proteins such as P65.

Reviewer #2 (Remarks to the Author):

In this manuscript, Luan and colleagues address the role of an NLR, NLRC5, in vascular remodeling. They find that its expression is increased in the neointima of injured arteries. Neointima formation is greatly enhanced in NLRC5-deficient mice, indicating that this NLR modulates this process. This is independent of NLRC5 expression in hematopoietic cells. Knockdown of NLRC5 in vascular smooth muscle cells promotes their expansion. NLRC5 is found to interact with PPARG, which has been reported to reduce vascular hyperplasia, and this interaction seems to favor PPARG activity. Furthermore, overexpressed PPARG is found to bind the promoter and its activity regulate NLRC5 expression.

The findings have strong and novel aspects; in particular, results in NLRC5 knockout are very impressive. Nonetheless, I have concerns with part of the data.

The major problem with this manuscript is that it fails to demonstrate the functional and physiological role of PPARg upstream and, in particular, downstream of NLRC5.

For the latter point, panels 4O and 4P address the consequences PPARg activation using an agonist on the expression of PCNA and CyclinD1 in the presence or absence of NLRC5. In these panels, NLRC5 silencing does not induce PCNA and CyclinD1, opposite to what shown and discussed for Figure 3E and F (unstimulated condition; “In our hand, knocking down NLRC5 resulted in downregulated α -SMA expression level with increased proliferative markers PCNA and Cyclin D1 expression, indicating that NLRC5 depletion promoted VSMC dedifferentiation, a process of switching from contractile phenotype to proliferation phenotype”). Rather, for PCNA, a reduction is observed; this raises questions on data reproducibility/interpretation. Does this work only downstream of PDGF? Then I would invite the authors to convey this message in a clearer form and explore the role of PDGF in vivo in control and NLRC5 knockouts.

Secondly, we lack information on cell proliferation and in vivo relevance of the NLRC5-PPARg interaction.

Last, activation of PPARg reduced PCNA and Cyclin D1 expression in PDGF-stimulated cells irrespectively of NLRC5 (a 50% reduction is observed in all conditions). Therefore, its activity does not seem to be regulated by NLRC5; rather, NLRC5 has a PPARg-independent activity in regulating the levels of PCNA and CyclinD1 upon PDGF treatment.

Minor points:

- Histology staining for NLRC5 in wt and ko tissues would have nicely confirmed the specificity of the staining.
- Supplementary Figure VIIA: labels of panel B are unclear.
- Figure 4 A: the pattern of NLRC5 in the IP seems to follow the one of NLRC5 expression, but whole cell lysates are not shown. These shall be shown and, ideally, the IP shall be performed in cells KD for PPARg (KD or KO approaches to study the role of PPARg would have been helpful in general to support the paper's conclusions).
- Control conditions in Figures 2B and 2H are extremely different.
- I would strongly suggest the authors to rework the text (grammar, conclusions, as well as depth and focus of the discussion).

- Experiment representativeness is not always mentioned in the figure legends.

Reviewer #3 (Remarks to the Author):

The manuscript titled “NLRC5 inhibits neointima formation following vascular injury through direct interaction with PPAR α ”, by P. Luan, et. al. demonstrates a novel function for the NOD-like receptor family member, NLRC5, on smooth muscle cell (SMC) function. The authors demonstrate that NLRC5 is expressed in the nucleus (and cytoplasm) of SMCs both in vivo in human coronary arteries and mouse carotid arteries and in vitro in cultured human SMCs; vascular injury and PDGF stimulation upregulate NLRC5 expression. Using forced loss- and gain-of-expression approaches (NLRC5 KO or adenoviral-mediated overexpressing mice; siRNA/adenoviral-mediated NLRC5 deficient/overexpressing SMCs), the authors demonstrate that NLRC5 plays a protective role against pathological vascular remodeling through decreased SMC proliferation and migration. Reciprocal bone marrow transplant studies demonstrated that exacerbated neointima formation in global NLRC5 KO mice is due to dysfunctional resident vascular cells and not recruited bone marrow-derived inflammatory cells. Finally, the authors demonstrate that NLRC5 interacts with the transcription factor PPAR α through the NACHT domain of NLRC5, interaction promotes PPAR α transcriptional activity, and in a feedback manner, PPAR α transcriptional activity regulates NLRC5 expression. This is an interesting and novel paper from the standpoint that while NOD-like receptor family member function is well studied in the regulation of inflammation, little is known of these proteins, and in particular NLRC5, in vascular biology. The authors used combined in vivo genetic approaches and in vitro molecular approaches to support their conclusions, which is a strength of the paper. There are several concerns, some major and some more minor, that the authors will need to address, as outlined below. Importantly, the authors appear to have a weak understanding of vascular biology and in particular SMC biology. Consulting with an investigator with expertise in this area would strengthen the manuscript.

Major

The data supporting a role for NLRC5 in repressing SMC proliferation and migration is good, but the data supporting its role in regulating SMC dedifferentiation, as characterized by decreased expression of SMC-specific differentiation genes is underwhelming. The Western for α SMA (Figure 3E) does not represent the data shown in the graph and α SMA is only one of several markers used to study SMC dedifferentiation. The authors should include additional markers in their analysis (e.g. smooth muscle myosin heavy chain, calponin). There is also confusing data presented. Figure 3 shows α SMA as a marker of dedifferentiation, but Sup Figure 7 shows Klf4 expression, although the text states α SMA (line 382). Please address the significance of Klf4 in this setting. Typically, PDGF induces Klf4, which promotes loss of SMC marker expression and dedifferentiation. The authors’

data oppose a large body of literature showing induction of Klf4 in the setting of vascular injury and/or PDGF stimulation. Finally, the statement “ α -smooth muscle actin (α SMA) has previously been reported as a key regulator of downstream VSMC differentiation gene expression” (lines 376-378) is incorrect. α SMA is a SMC differentiation-associated contractile gene, not a transcriptional regulator.

The mechanistic data to support a NLRC5-PPAR α axis in the regulation of SMC phenotype was not forcefully addressed and some of the data do not support the conclusions. Many of the representative Westerns do not accurately reflect the quantitative data (Figures 3&4, Sup Figure 7). Figure 4P – if pioglitazone-PPAR α are working through NLRC5 induction to blunt protein expression, why is the effect of pioglitazone not lost in NLRC5-deficient SMCs?

The wording regarding overall results is sometimes confusing. The data implicate NLRC5 as being protective against pathological vascular remodeling, yet the opening discussion reads that it promotes vascular remodeling (lines 499-503). The authors previous work demonstrated that loss of NLRC5 blocked diabetic nephropathy and it is referred that this is a disease where vascular remodeling is highly involved suggesting that loss of NLRC5 would block vascular remodeling as well. The current findings are not in agreement with this suggestion as loss of NLRC5 makes remodeling worse. In addition, please address the significance of induction of NLRC5 in the setting of disease/vascular injury, especially in light of its protective role in blocking remodeling.

In siRNA-mediated KO studies, there appears to be a basal increase in many of the readouts in the absence of PDGF stimulation (proliferation, migration, proliferative marker expression). Please address the significance of these findings. In addition, although the authors state that ANOVA was used to analyze multiple groups, the statistics shown do not seem to support this type of analysis. Also, the basal, non-stimulated increase in PCNA and Cyclin D1 expression observed in NLRC5-deficient SMCs (Figure 3) appears to be lost in Figure 4.

Nuclear localization of NLRC5 in SMCs was overinterpreted and incorrect. In in vivo images (Figure 1B&D, Sup Figure 2) and in vitro images (Figures 1J, 4B&D) there is clearly cytoplasmic localization (there would be no yellow staining in overlay images with NLRC5 and α SMA staining if NLRC5 was strictly nuclear as α SMA is restricted to the cytoplasm). Please alter the text to reflect this (in particular lines 568-569).

The number of human samples analyzed is very low and only N=1 from normal vs. atherosclerotic was shown. The representative atherosclerotic images in the immunofluorescent images is not representative as there is very little intimal hyperplasia (this image is more representative of a normal human coronary vessel). Please include example confocal images from all stained human

vessels in a supplementary figure. In addition, as the authors referred to “the proliferated medial layer...” (line 293), triple staining for NLRC5, α SMA, and PCNA would strengthen the data.

Minor

p.20, lines 439-444: the text is confusing as the authors first refer to pioglitazone stimulation, then change to PDGF-BB stimulation. Please address and correct.

Figure 5 and text: Please address the significance and define the various NLRC5 domains.

Figure 2: Please show higher magnification images of PCNA-stained arteries to better appreciate the data (similar to Figure 6F).

Figure 4A: Please show reciprocal IP (IP NLRC5, IB PPAR α).

Figure 4F and Figure 5E-G: It is difficult to appreciate a dose response of overexpression making the robust response at the highest dose questionable. Please address.

More detail should be included in the figure legends (e.g. number of independent experiments/westerns conducted; quantification of NLRC5-, PCNA-positive cells)

Are the data in Sup Figure IV significant for CD8+ T-cells? If so, please indicate on the figure.

Response to Reviewers

Reviewer #1 (Remarks to the Author):

This is a very interesting paper describing a role for NLRC5 as an inhibitor of neointima formation through an interaction with PPAR γ . The data and implications are highly significant and novel. The paper would be improved by some additional data flushing out mechanisms of the NLRC5/PPAR γ pathway as defined in the comments below.

Reply: We highly appreciate your careful review and constructive suggestions. To make our data more reliable, we presented the immunofluorescence staining of Flag and confirmed the efficiency of NLRC5 knockdown in vitro. To dissect and validate the feedback between NLRC5 and PPAR γ in vivo and in vitro, we administrated PPAR γ agonist pioglitazone and performed a series of immunoprecipitation followed with immunoblots, PCR and morphological analysis. The specific questions are now replied point-by-point as follows.

Major Concerns

1. Given the large, easily detectable increase in *Nlrc5* by western blot (Figure 2G), it would be attractive to demonstrate whether this was expressed in endothelial cells, smooth muscle cells or adventitia after the gene transfer experiment. The use of the FLAG tag should easily facilitate this.

Reply: Thank you for your suggestion. By immunofluorescence staining, we found that adenoviruses labeled with Flag tag in green co-localized with smooth muscle cells labeled with α -SMA and endothelial cells labeled with CD31. We have updated this figure in Supplementary Figure IX.

On page 17,

Overexpression of *Nlrc5* was verified by both western blot analyses and immunofluorescence staining (Figure 2G and Supplementary Figure IX).

2. The efficacy of the siNLRC5 in Figure 4E is unclear. Can you quantify the knock down in each sample used to measure Luc activity?

Reply: We performed western blot analyses to validate the efficiency of siRNA-mediated knockdown of NLRC5 in HEK293T cells. These data are presented in Figure 4.

3. Page 20- you say “The ChIP-qPCR results showed that PPAR γ -binding to NLRC5 promoter greatly increased in response to PDGF-BB stimulation (Figure 4L)”. Figure 4L does not show the ChIP response to PDGF, it shows response to Pioglitazone.

Reply: We have acknowledged this error and updated accordingly. To confirm the effect of PPAR γ on NLRC5 transcription, we performed ChIP-qPCR on NLRC5 promoter in response to PPAR γ agonist Pioglitazone.

On page 24,

The ChIP-qPCR studies showed that the PPAR γ -binding to the NLRC5 promoter significantly increased in response to Pioglitazone (10 nM) (Figure 5H), further supporting our hypothesis.

4. You are suggesting a forward feedback mechanism whereby PPARG activation increases PPARG-mediated transcription of NLRC5 which then binds to PPARG further facilitating PPARG transcriptional activity. A) Is there evidence for the induction of bona fide PPARG target genes in HASMC that is mediated by NLRC5/PPARG complex? B) Does the complex require a PPARG ligand to form? C) Does NLRC5 participate in the PPARG/PPRE complex? D) Does NLRC5 act as a PPARG ligand?

Reply: A) According to previous publications (1-3), some PPAR γ target genes related to proliferation and migration, such as CD36, AP2 and CITED2, were induced. To confirm the role of NLRC5 and NLRC5/PPAR γ on PPAR γ target genes, we knocked down NLRC5 and used quantitative PCR to determine the expression of these candidate genes.

On page 21,

To confirm effects of NLRC5-mediated regulation of the PPAR γ transcriptional network, we examined the expression of well-known PPAR γ target genes related to

proliferation and migration, including CD36, AP2 and CITED2 (1-3). In response to siRNA-mediated *NLRC5* knockdown, the mRNA expressions of CD36, AP2, and CITED2 were significantly decreased (Figure 4I)

1

References

1. Duan, S. Z., Usher, M. G., and Mortensen, R. M. (2008) Peroxisome proliferator-activated receptor-gamma-mediated effects in the vasculature. *Circ Res* **102**, 283-294
2. Tung, E. W. Y., Ahmed, S., Peshdary, V., and Atlas, E. (2017) Firemaster(R) 550 and its components isopropylated triphenyl phosphate and triphenyl phosphate enhance adipogenesis and transcriptional activity of peroxisome proliferator activated receptor (Ppargamma) on the adipocyte protein 2 (aP2) promoter. *PLoS One* **12**, e0175855
3. Cheung, K. F., Zhao, J., Hao, Y., Li, X., Lowe, A. W., Cheng, A. S., Sung, J. J., and Yu, J. (2013) CITED2 is a novel direct effector of peroxisome proliferator-activated receptor gamma in suppressing hepatocellular carcinoma cell growth. *Cancer* **119**, 1217-1226

B) To address this issue, we add pioglitazone that serves as an exogenous PPAR γ ligand and performed IP experiments. We find that treatment of pioglitazone enhanced the expression of *NLRC5* and the interaction of *NLRC5*/PPAR γ . We describe these observations and incorporated this important information in Figure 5E.

On page 24,

As an initial step, when HASMCs were treated with pioglitazone (10 nM), the mRNA and protein expression levels of *NLRC5* were gradually increased in a time-dependent manner (Figure 5C and 5D). Notably, while the intrinsic interaction between *NLRC5* and PPAR γ existed without stimulation of the exogenous PPAR γ ligand, treatment of pioglitazone further enhanced the interaction of *NLRC5*/PPAR γ (Figure 5E).

C) PPAR γ functions through formation of an obligate heterodimer with retinoid X receptors α (RXR α). PPAR γ /RXR α heterodimers bind to the PPAR response element (PPREs) in the regulatory regions of target genes and activate gene transcription. We therefore knocked down NLRC5 and performed IP to determine whether NLRC5 acted as a ligand and mediated PPAR γ /RXR α complex formation. However, our results showed that the PPAR γ /RXR α complex formation was independent of NLRC5.

On page 22,

Given that PPAR γ /retinoid X receptors α (RXR α) heterodimers bind to PPRE in the regulatory regions of target genes and activate gene transcription, we performed IP to determine whether the presence of NLRC5 regulated PPAR γ /RXR α complex formation. However, absence of NLRC5 did not alter the PPAR γ -RXR α interaction in HASMCs (Figure 4J).

D) The above results confirmed that NLRC5 act as a PPARG ligand?

Reply: To further confirm that NLRC5 functions as a PPAR γ ligand without interfering with the PPAR γ /RXR α complex, we generated a plasmid carrying the LBD of PPAR γ and determined that the LBD domain, which was responsible for the binding of PPAR γ ligand, was required for the interaction with NLRC5.

On page 22,

To dissect whether NLRC5 serves as a PPAR γ ligand without interfering with the PPAR γ /RXR α complex, we generated a Flag-PPAR γ LBD and determined the intrinsic interaction between NLRC5 and LBD of PPAR γ (Figure 4K). Co-immunoprecipitation revealed the interaction of NLRC5 with the LBD of PPAR γ , suggesting that the LBD domain, which is responsible for the binding of PPAR γ ligand, is required for the interaction with NLRC5 (Figure 4L). These experiments indicate that NLRC5 predominantly interacts with PPAR γ , acts as a PPAR γ ligand and regulates PPARE-dependent gene transcription without alteration of the PPAR γ /RXR α complex.

5. The experiment in Figure 4O-P concluding that NLRC5 protected vascular remodeling through a positive feedback loop with PPAR γ in VSMCs would be much stronger if it was coupled to an experiment using a PPAR γ antagonist or siRNA targeting PPAR γ . This would show that the protective effects of NLRC5 are mediated by, or require PPAR γ .

Reply: Thank you for your suggestion. We further added the PPAR γ antagonist T0070907 to counteract the protective effect of NLRC5 on VSMC proliferation and dedifferentiation. The results showed that T0070907 counteracted the NLRC5 effects on VSMCs.

On page 23,

Conversely, the enhancement of α -SMA, Calponin, and Myosin expression in HASMCs transduced with Ad-NLRC5 was blocked in the presence of the PPAR γ antagonist T0070907, and accompanied by a deterioration of PCNA and Cyclin D1 expression. These findings indicate that T0070907 counteracted the protective effect of NLRC5 on VSMC proliferation and dedifferentiation (Supplementary Figure XVB and XVC).

6. The data in Figure 5 elegantly show that the NACHT domain is all that is required to bind to PPARG and mediate an induction in PPARG activity. It would be attractive to confirm this effect on a bona fide PPARG target gene, perhaps CD36 or AP2, the latter is ubiquitously expressed in many cell types and is quite abundant in SMC.

Reply: As suggested, we transfected HASMCs with NACHT domain overexpression adenovirus and performed quantitative PCR to evaluate the expression of PPARG target genes. The data showed that overexpression of the NACHT domain led to increased expression of PPARG target genes.

On page 26,

Furthermore, after adenoviral overexpression of the NACHT domain of NLRC5 (Ad-NLRC5 NACHT), the expression of the PPARG target genes CD36, AP2, and CITED2 were markedly higher than those measured in Ad-Ctr-transduced HASMCs (Figure 6K).

7. Like you showed in Figure 2H-L, is over-expression of the NACHT domain sufficient to mediate a decrease in intimal area and HASMC proliferation?

Reply: After localized delivery of adenovirus bearing the NACHT domain in ligated carotids, we performed H&E and immunohistochemistry to assess intima formation and proliferation. Furthermore, we transduced HASMCs with an adenovirus

overexpressing the NACHT domain and performed Edu to validate in vitro proliferation. The data has now been incorporated in the results section.

On page 25,

We then administrated adenovirus containing the NACHT domain of *NLRC5* (Ad-*NLRC5* NACHT) or empty adenovirus (Ad-Ctr) in HASMCs and assessed the effect of the NACHT domain on VSMC proliferation. The Edu incorporation assay showed that adenoviral overexpression of the NACHT domain substantially attenuated VSMC proliferation compared with Ad-Ctr groups (Figure 6I and 6J).

On page 26,

To assess whether the suppressive role of the NACHT domain on VSMC proliferation in vitro similarly regulates neointimal formation in vivo, we ligated common carotid arteries in mice and performed local transduction of Ad-Control or Ad-*Nlrc5* NACHT. Immunoblots for Flag tag were performed and the efficiency of localized delivery of adenovirus into carotid arteries was confirmed (Figure 6L). In accordance with the in vitro observations, the cross-sectional areas of neointima (Ad-Control $1.29 \pm 0.38 \times 10^4 \mu\text{m}^2$ vs. Ad-*Nlrc5* NACHT $0.48 \pm 0.17 \times 10^4 \mu\text{m}^2$, $P < 0.01$) and intima/media ratios (Ad-Control 0.60 ± 0.11 vs. Ad-*Nlrc5* NACHT 0.32 ± 0.07 , $P < 0.01$) in ligated carotid arteries were markedly increased in Ad-*Nlrc5* NACHT mice compared with Ad-Control mice (Figure 6M-O). Immunohistochemistry analysis of PCNA-positive cells within ligated carotid arteries indicated that VSMC proliferation of Ad-Control mice were more severe than that of Ad-*Nlrc5* NACHT mice (Figure 6P and 6Q).

Minor Concerns

8. It would be preferred to perform BP measurements with an assay which is more reliable than tail cuff as there appears to be a trend toward an increase in the KO mice (Figure S3A). If showing tail cuff, only show the SBP as the DBP is a calculated variable not a measured variable.

Reply: Thank you for your suggestion. We repeated the experiments as follows: For systolic blood pressure (BP) monitoring, mice were acclimated for at least 3 consecutive days before BP measurements and for one hour prior to performing the experiment. Then, all mice were encouraged to walk into the restraining tube, which was adjusted to prevent excessive movement. BP and heart rate measurement were carried out at the same time in a predetermined quiet area. BP measurements were taken three times from each mouse, and the average number was reported in this study.

On page 16,

We also measured systolic BP, heart rate, and analyzed the lipid profile to exclude potential confounding factors associated with vascular remodeling. Tail-cuff BP measurement showed no difference in systolic BP (*Nlrc5*^{+/+} 113.68 ± 9.45 mmHg vs.

Nlrc5^{-/-} 116.09 ± 11.46 mmHg) and heart rate between *Nlrc5*^{-/-} and *Nlrc5*^{+/+} mice after 3-week complete carotid ligation (Supplementary Figure VA and VB).

9. Please check if the correct blots for p-IκB and IκB are used in Figure S9A as they look identical and there does not appear to be induction of P-IκB by PDGF.

Reply: We had repeated the western blots for p-IκBα and t-IκBα expression to confirm the data, and found that the incubation with PDGF-BB moderately stimulated IκBα phosphorylation, while knockdown of NLRC5 did not affect the phosphorylation expression levels of IκBα. We have replaced the Supplementary Figure XII with this figure.

10. Whenever possible, it would be more attractive to use increased transparency in the reporting of data by substituting bar/plunger graphs with dot/whisker plots.

Reply: As suggested, we substituted some statistical graphs, especially morphology analysis, with scatter plots.

For example,
Figure 2H-L

11. Discussion: It would be worthwhile citing other studies showing the direct association between PPARG and other proteins such as P65.

Reply: We were glad to receive such constructive comments. As suggested, we cited relevant literature and discussed the association of PPAR γ with other co-activators and/or transcription factors.

On page 31,

It is known that PPAR γ functions with the obligate heterodimer RXR α . PPAR γ /RXR α heterodimers binds to PPRE in the regulatory regions of target genes and activate gene transcription. Apart from this canonical PPAR γ receptor, Zhang et al. revealed that NF- κ B p65 subunit and PPAR γ formed an active transcription factor complex and cooperatively inhibited atherosclerotic progression. Nevertheless, our results show that absence of NLRC5 does not disrupt PPAR γ /RXR α heterodimer formation, but suppresses the transcriptional activity of downstream genes. Furthermore, the observations that the LBD domain of PPAR γ directly interacted with NLRC5, together with the in vivo and in vitro rescue experiments using PPAR γ agonist pioglitazone, suggest that NLRC5 may serve as an endogenous ligand for PPAR γ and provides a protective shield against vascular injury via activation of PPAR γ .

Reviewer #2 (Remarks to the Author):

In this manuscript, Luan and colleagues address the role of an NLR, NLRC5, in vascular remodeling. They find that its expression is increased in the neointima of injured arteries. Neointima formation is greatly enhanced in NLRC5-deficient mice, indicating that this NLR modulates this process. This is independent of NLRC5 expression in hematopoietic cells. Knockdown of NLRC5 in vascular smooth muscle cells promotes their expansion. NLRC5 is found to interact with PPAR γ , which has been reported to reduce vascular hyperplasia, and this interaction seems to favor PPAR γ activity. Furthermore, overexpressed PPAR γ is found to bind the promoter and its activity regulate NLRC5 expression.

Reply: Thank you for your critical suggestion. First, we acknowledge the confusing results relevant to the reciprocal effects of NLRC5 and PPAR γ . To dissolve these issues, we administrated PPAR γ agonist pioglitazone in WT/KO mice and in HASMC transfected with NLRC5 siRNA, finding that recovery of PPAR γ activity partly counteracted the deterioration of NLRC5 depletion on neointimal formation and VSMC proliferation and dedifferentiation. We also tried our best to verify the interaction between NLRC5 and PPAR γ in vivo and in vitro. Collectively, these data supported the relevance and role of NLRC5-PPAR γ interaction in VSMCs. The specific questions are now replied point-by-point as follows.

The findings have strong and novel aspects; in particular, results in NLRC5 knockout are very impressive. Nonetheless, I have concerns with part of the data.

1. The major problem with this manuscript is that it fails to demonstrate the functional and physiological role of PPAR γ upstream and, in particular, downstream of NLRC5. For the latter point, panels 4O and 4P address the consequences PPAR γ activation using an agonist on the expression of PCNA and CyclinD1 in the presence or absence of NLRC5. In these panels, NLRC5 silencing does not induce PCNA and CyclinD1, opposite to what shown and discussed for Figure 3E and F (unstimulated condition; “In our hand, knocking down NLRC5 resulted in downregulated α -SMA expression level with increased proliferative markers PCNA and Cyclin D1 expression, indicating that NLRC5 depletion promoted VSMC dedifferentiation, a process of switching from contractile phenotype to proliferation phenotype”). Rather, for PCNA, a reduction is observed; this raises questions on data reproducibility/interpretation. Does this work only downstream of PDGF? Then I would invite the authors to convey this message in a clearer form and explore the role of PDGF in vivo in control and NLRC5 knockouts.

Reply: We acknowledge the confusing results relevant to the reciprocal effects of NLRC5 and PPAR γ on PCNA, CyclinD1, and differentiation markers. To clarify this issue, we optimized the experimental conditions including the concentration and duration of PPAR γ ligand stimulation. Additionally, to interpret the confusing data and verify reproducibility, we repeated the western blot analyses. The data showed that knockdown of NLRC5 significantly enhanced the expression of PCNA and Cyclin D1 in the presence of PDGF stimulation.

On page 18,

Consistent with the *in vivo* findings, *NLRC5* knockdown reduced the expression of α -SMA, Calponin, and Myosin concomitant with increased expression of proliferative markers PCNA and Cyclin D1. These data indicate that *NLRC5* depletion promoted VSMC dedifferentiation, a process of switching from a contractile to proliferative phenotype.

On page 23,

PPAR γ agonist pioglitazone treatment led to a significant reduction in PCNA and Cyclin D1 expression, accompanied with a recovery of α -SMA, Calponin, and Myosin expression, in the presence and absence of *NLRC5* (Figure 4R and Supplementary Figure XVA).

2. Secondly, we lack information on cell proliferation and *in vivo* relevance of the *NLRC5*-PPAR γ interaction.

Reply: To address this, we performed immunohistochemistry staining to confirm the co-localization of NLRC5 and PPAR γ within carotid arteries. Furthermore, we administrated pioglitazone to *Nlrc5*^{+/+} and *Nlrc5*^{-/-} mice, aiming to determine whether PPAR γ agonist pioglitazone rescues the excessive neointimal formation observed in *Nlrc5*^{-/-} mice and explain the role of NLRC5-PPAR γ interaction in vivo.

On page 21,

In parallel, we found that *Nlrc5* expression substantially co-localized with PPAR γ in serial sections of mouse carotid arteries after 1-week of carotid ligation (Figure 4C).

On page 23,

Second, we found that treatment of pioglitazone only partly rescued the excessive neointimal formation in *Nlrc5*^{-/-} mice as compared to *Nlrc5*^{-/-} mice without pioglitazone treatment (Figure 4M-O). Concomitantly, pioglitazone significantly alleviated the *in vivo* proliferation as quantified by PCNA staining in ligated carotid arteries from *Nlrc5*^{-/-} mice (Figure 4P and 4Q).

3. Last, activation of PPAR γ reduced PCNA and Cyclin D1 expression in PDGF-stimulated cells irrespectively of NLRC5 (a 50% reduction is observed in all conditions). Therefore, its activity does not seem to be regulated by NLRC5; rather, NLRC5 has a PPAR γ -independent activity in regulating the levels of PCNA and CyclinD1 upon PDGF treatment.

Reply: Thank you for your suggestion. To clarify the role of NLRC5 in PPAR γ signaling, HASMCs were transfected with control siRNA or NLRC5 siRNA followed

by pioglitazone treatment. We found that pioglitazone partly rescued the increased PCNA and Cyclin D1 expression in the presence and absence of NLRC5.

On page 23,

To examine whether PPAR γ contributed to NLRC5-mediated alleviation of VSMC proliferation and dedifferentiation *in vitro*, HASMCs were treated with PPAR γ agonist pioglitazone (10 nM) with and without NLRC5 depletion. PPAR γ agonist pioglitazone treatment led to a significant reduction in PCNA and Cyclin D1 expression, accompanied with a recovery of α -SMA, Calponin, and Myosin expression, in the presence and absence of NLRC5 (Figure 4R and Supplementary Figure XVA).

Minor points:

4. Histology staining for NLRC5 in wt and ko tissues would have nicely confirmed the specificity of the staining.

Reply: As suggested, we performed immunofluorescence staining in ligated carotid arteries from WT and KO mice to confirm the specificity of NLRC5 staining.

On page 16,

To investigate the effect of NLRC5 on neointimal formation, *Nlrc5*^{-/-} and *Nlrc5*^{+/+} mice were subjected to vascular injury by carotid ligation for 3 weeks. We first verified the success of *Nlrc5* deletion in *Nlrc5*^{-/-} mice and tested the specificity of

Nlrc5 staining (Supplementary Figure IV).

5. Supplementary Figure VIIA: labels of panel B are unclear.

Reply: We appreciate your careful review and have updated the figure labels.

6. Figure 4 A: the pattern of NLRC5 in the IP seems to follow the one of NLRC5 expression, but whole cell lysates are not shown. These shall be shown and, ideally, the IP shall be performed in cells KD for PPAR γ (KD or KO approaches to study the role of PPAR γ would have been helpful in general to support the paper's conclusions).

Reply: This is an important suggestion. We performed western blots for the expression of NLRC5 and PPAR γ in whole cell lysates. Additionally, we supplemented the experiments that involved NLRC5 knockdown by siRNA and elucidated the interaction between NLRC5 and PPAR γ by IP.

On page 21,

PDGF-BB stimulation (10 ng/ml) remarkably promoted the intrinsic interaction of NLRC5 with PPAR γ in HASMCs within 6-12 hours (Figure 4A). Vice versa, immunoprecipitation with NLRC5 antibodies followed by immunoblot analyses showed an increase in the interaction of NLRC5 with PPAR γ in the presence of PDGF-BB (Figure 4A).

On page 24,

Moreover, transient transfection with PPAR γ siRNA in HASMCs and co-immunoprecipitation followed by immunoblot analyses demonstrated that

knockdown of PPAR γ reduced NLRC5 expression and abrogated the interaction between NLRC5 and PPAR γ as well (Figure 5F).

7. Control conditions in Figures 2B and 2H are extremely different.

Reply: We thank the reviewer for these points regarding the H&E images in Figure 2H. As suggested, we changed the representative images in Figure 2H.

Figure 2H

8. I would strongly suggest the authors to rework the text (grammar, conclusions, as well as depth and focus of the discussion).

Reply: We thank the reviewer for this suggestion. Throughout the manuscript, we had checked for grammar errors. We have also incorporated more depth and breadth in the discussion section.

9. Experiment representativeness is not always mentioned in the figure legends.

Reply: The figure legends have been updated accordingly.

Reviewer #3 (Remarks to the Author):

The manuscript titled “NLRC5 inhibits neointima formation following vascular injury through direct interaction with PPAR”, by P. Luan, et. al. demonstrates a novel function for the NOD-like receptor family member, NLRC5, on smooth muscle cell (SMC) function. The authors demonstrate that NLRC5 is expressed in the nucleus (and cytoplasm) of SMCs both in vivo in human coronary arteries and mouse carotid arteries and in vitro in cultured human SMCs; vascular injury and PDGF stimulation upregulate NLRC5 expression. Using forced loss- and gain-of-expression approaches (NLRC5 KO or adenoviral-mediated overexpressing mice; siRNA/adenoviral-mediated NLRC5 deficient/overexpressing SMCs), the authors demonstrate that NLRC5 plays a protective role against pathological vascular remodeling through decreased SMC proliferation and migration. Reciprocal bone marrow transplant studies demonstrated that exacerbated neointima formation in global NLRC5 KO mice is due to dysfunctional resident vascular cells and not recruited bone marrow-derived inflammatory cells. Finally, the authors demonstrate that NLRC5 interacts with the transcription factor PPAR through the NACHT domain of NLRC5, interaction promotes PPAR transcriptional activity, and in a feedback manner, PPAR transcriptional activity regulates NLRC5 expression. This is an interesting and novel paper from the standpoint that while NOD-like receptor family member function is well studied in the regulation of inflammation, little is known of these proteins, and in particular NLRC5, in vascular biology. The authors used combined in vivo genetic approaches and in vitro molecular approaches to support their conclusions, which is a strength of the paper. There are several concerns, some major and some more minor, that the authors will need to address, as outlined below. Importantly, the authors appear to have a weak understanding of vascular biology and in particular SMC biology. Consulting with an investigator with expertise in this area would strengthen the manuscript.

Reply: We highly appreciate your comments. First of all, we acknowledge that our understanding of vascular biology is limited. To strengthen the weakness in our work, Prof. Mark from Brigham and Women’s Hospital, Harvard Medical School is invited to help design the study and conduct the revision work. We have included other VSMC dedifferentiation markers in the following experiments and rephrased irrelevant expression in the text. Indeed, we observed that protective effect of PPAR γ agonist pioglitazone on VSMC proliferation retained in NLRC5-deficient VSMCs, reflecting that the function of pioglitazone on VSMCs did not totally rely on its interaction with NLRC5. The specific questions are now replied point-by-point as follows.

Major concerns:

1. The data supporting a role for NLRC5 in repressing SMC proliferation and migration is good, but the data supporting its role in regulating SMC dedifferentiation, as characterized by decreased expression of SMC-specific differentiation genes is underwhelming. The Western for α SMA (Figure 3E) does not represent the data shown in the graph and α SMA is only one of several markers used to study SMC

dedifferentiation. The authors should include additional markers in their analysis (e.g. smooth muscle myosin heavy chain, calponin).

Reply: We agreed with the reviewer and had supplemented the western blots to explore the expression of other VSMC dedifferentiation markers. Since the molecular weights of β -actin and GAPDH approximated those of α -SMA and Calponin, Vinculin was used as an appropriate substitute for the above house-keeping genes. We added these data in the results section and Figure 3.

On page 18,

Accumulating studies highlight that contractile VSMCs are capable of dedifferentiating into synthetic VSMCs in response to vascular injury and several extracellular stimuli. This process is called VSMC phenotype switching, which coordinates with VSMC proliferation and migration. Myosin, α -SMA, and Calponin have previously been reported as key regulators of downstream VSMC differentiation gene expression (4). Consistent with the *in vivo* findings, *NLRC5* knockdown reduced the expression of α -SMA, Calponin, and Myosin concomitant with increased expression of proliferative markers PCNA and Cyclin D1. These data indicate that *NLRC5* depletion promoted VSMC dedifferentiation, a process of switching from a contractile to proliferative phenotype (Figure 3E and 3F).

Reference

4. Cheng, Y., Liu, X., Yang, J., Lin, Y., Xu, D. Z., Lu, Q., Deitch, E. A., Huo, Y., Delphin, E. S., and Zhang, C. (2009) MicroRNA-145, a novel smooth muscle cell phenotypic marker and modulator, controls vascular neointimal lesion formation. *Circulation research* **105**, 158-166

On page 19,

In contrast, *NLRC5* overexpression led to increased expression of α -SMA, Calponin, and Myosin, but decreased expression of PCNA and Cyclin D1 in HASMCs upon PDGF-BB stimulation (10 ng/ml) (Supplementary Figure XD and XE).

2. There is also confusing data presented. Figure 3 shows α SMA as a marker of dedifferentiation, but Sup Figure 7 shows Klf4 expression, although the text states α SMA (line 382). Please address the significance of Klf4 in this setting. Typically, PDGF induces Klf4, which promotes loss of SMC marker expression and dedifferentiation. The authors' data oppose a large body of literature showing induction of Klf4 in the setting of vascular injury and/or PDGF stimulation.

Reply: We thank the reviewer for these important insights. We acknowledged this discrepancy, as we did not further explore the effect of NLRC5 on Klf4 in VSMCs. In our preliminary experiments, we found that NLRC5 strongly associated with PPAR γ , whereas it more modestly associated with Klf4 by western blot analyses. Therefore, based upon our data on PPAR γ and previous literature related to the function of PPAR γ in vascular remodeling, we focused this study on establishing the role of NLRC5 on PPAR γ in subsequent experiments. We had deleted the confusing data relevant to Klf4 in order to avoid misunderstanding by reviewers and readers.

3. Finally, the statement " α -smooth muscle actin (α SMA) has previously been reported as a key regulator of downstream VSMC differentiation gene expression" (lines 376-378) is incorrect. α -SMA is a SMC differentiation-associated contractile gene, not a transcriptional regulator.

Reply: We agree with the reviewer and have updated this sentence in our paper.

4. The mechanistic data to support a NLRC5-PPAR γ axis in the regulation of SMC phenotype was not forcefully addressed and some of the data do not support the conclusions. Many of the representative Westerns do not accurately reflect the quantitative data (Figures 3&4, Sup Figure 7). Figure 4P – if pioglitazone-PPAR γ are working through NLRC5 induction to blunt protein expression, why is the effect of pioglitazone not lost in NLRC5-deficient SMCs?

Reply: Thank you for these suggestions. We acknowledge that some data are not strong enough to support that NLRC5-PPAR γ axis in the regulation of SMC phenotype. To address this, we first constructed the plasmid carrying the ligand-binding domain (LBD) of PPAR γ and determined the intrinsic interaction between NLRC5 and LBD of PPAR γ (Figure 4K). Co-immunoprecipitation revealed the interaction of NLRC5 with the LBD of PPAR γ (Figure 4L).

On page 22,

To dissect whether NLRC5 serves as a PPAR γ ligand without interfering with the PPAR γ /RXR α complex, we generated a Flag-PPAR γ LBD and determined the intrinsic interaction between NLRC5 and LBD of PPAR γ (Figure 4K). Co-immunoprecipitation revealed the interaction of NLRC5 with the LBD of PPAR γ , suggesting that the LBD domain, which is responsible for the binding of PPAR γ ligand, is required for the interaction with NLRC5 (Figure 4L). These experiments indicate that NLRC5 predominantly interacts with PPAR γ , acts as a PPAR γ ligand and regulates PPRE-dependent gene transcription without alteration of the PPAR γ /RXR α complex.

Second, we found that treatment of pioglitazone only partly rescued the excessive neointimal formation in *Nlrc5*^{-/-} mice as compared to *Nlrc5*^{-/-} mice without pioglitazone treatment (Figure 4M-O). Concomitantly, pioglitazone significantly alleviated the *in vivo* proliferation labeled as quantified by PCNA staining in ligated carotid arteries from *Nlrc5*^{-/-} mice (Figure 4P and 4Q). This implied that the effect on SMC proliferation by pioglitazone-PPAR γ was dependent in part on the presence of NLRC5 *in vivo*.

On page 23,

Second, we found that treatment of pioglitazone only partly rescued the excessive neointimal formation in *Nlrc5*^{-/-} mice as compared to *Nlrc5*^{-/-} mice without pioglitazone treatment (Figure 4M-O). Concomitantly, pioglitazone significantly alleviated the *in vivo* proliferation as quantified by PCNA staining in ligated carotid arteries from *Nlrc5*^{-/-} mice (Figure 4P and 4Q).

Also, the aforementioned *in vitro* experiment showed that the effects on markers of SMCs by pioglitazone-PPAR γ were decreased in response to NLRC5 siRNA-mediated knockdown. Collectively, these findings support that pioglitazone-PPAR γ signaling through NLRC5 regulates SMC phenotype.

We not only replicated these western blot experiments, but also added an IP experiment to further explore the NLRC5-PPAR γ interaction. We find that interruption of the interaction with NLRC5 significantly enhanced the expression of PCNA and Cyclin D1 in response to PDGF stimulation.

On page 23,

To examine whether PPAR γ contributed to NLRC5-mediated alleviation of VSMC proliferation and dedifferentiation *in vitro*, HASMCs were treated with PPAR γ agonist pioglitazone (10 nM) with and without *NLRC5* depletion. PPAR γ agonist pioglitazone treatment led to a significant reduction in PCNA and Cyclin D1 expression, accompanied with a recovery of α -SMA, Calponin, and Myosin expression, in the presence and absence of NLRC5 (Figure 4R and Supplementary Figure XVA).

5. The wording regarding overall results is sometimes confusing. A. The data implicate NLRC5 as being protective against pathological vascular remodeling, yet the opening discussion reads that it promotes vascular remodeling (lines 499-503). B. The authors previous work demonstrated that loss of NLRC5 blocked diabetic nephropathy and it is referred that this is a disease where vascular remodeling is highly involved suggesting that loss of NLRC5 would block vascular remodeling as well. The current findings are not in agreement with this suggestion as loss of NLRC5 makes remodeling worse. C. In addition, please address the significance of induction of NLRC5 in the setting of disease/vascular injury, especially in light of its protective role in blocking remodeling.

Reply: We are appreciative of these insightful suggestions. We acknowledge that some of these statements appear potentially antithetical; however, the phenotype and molecular mechanisms of NLRC5 in vascular remodeling is indeed consistent through most of the text. To clarify, although NLRC5 was increased in the intimal layer in response to carotid ligation and in VSMCs when stimulated by PDGF-BB, we found that this increase counteracts the vascular remodeling. On the other hand, *in vivo* and *in vitro* experiments confirmed that knockdown of NLRC5 promoted vascular remodeling. We therefore revised the corresponding part in the discussion section to improve clarity.

As for the role of NLRC5 in diabetic nephropathy, we hypothesized that different environments or stimuli was the reason why NLRC5 aggravated the development of diabetic nephropathy. To prove this, we stimulated VSMCs with high glucose or PDGF-BB, and found that while PDGF-BB did not activate Smad2 phosphorylation, high glucose significantly stimulated Smad2 phosphorylation. These effects were consistent with analogous findings in mesangial cells as described in our previous work. Moreover, we determined that the expression of Myosin and α -SMA expression in response to diabetic nephropathy (DN) in different genetic mice. We found that the expression of VSMC markers remained unchanged in DN of *Nlrc5*^{+/+} and *Nlrc5*^{-/-} mice. These data have been presented in the results section and supplementary figures.

On page 20,

In contrast to the protective effect of NLRC5 in vascular remodeling, our prior study found that NLRC5 played a contradictory role in diabetic mice that loss of NLRC5 ameliorated diabetic nephropathy (DN). Therefore, we hypothesized that different environments or stimuli may influence the function of NLRC5. To prove this, we

stimulated VSMCs with high glucose or PDGF-BB, and found that while PDGF-BB did not activate Smad2 phosphorylation, high glucose significantly stimulated Smad2 phosphorylation (Supplementary Figure XIII A-D). These effects are consistent with analogous findings in mesangial cells as described in our previous work. Moreover, we determined the expression of Myosin, α -SMA, Cyclin D1 and PCNA expression in response to DN in different genetic mice. We found that the expression of proliferation and VSMC markers remained unchanged in kidneys of *Nlrc5*^{+/+} diabetic mice compared with *Nlrc5*^{-/-} diabetic mice (Supplementary Figure XIII E and F).

6. In siRNA-mediated KO studies, A. there appears to be a basal increase in many of the readouts in the absence of PDGF stimulation (proliferation, migration, proliferative marker expression). Please address the significance of these findings. B. In addition, although the authors state that ANOVA was used to analyze multiple groups, the statistics shown do not seem to support this type of analysis. C. Also, the basal, non-stimulated increase in PCNA and Cyclin D1 expression observed in NLRC5-deficient SMCs (Figure 3) appears to be lost in Figure 4.

Reply: We agree with the reviewer and acknowledge that some of the relevant data are inconsistent and confusing. Therefore, we had repeated the western blots and performed statistical analysis based on one-way ANOVA and post-hoc Bonferroni test between-groups as demonstrated in the methods section. To improve clarity of the data, we illustrated the statistical methods in Figure legends when necessary. Additionally, we found that either depletion or overexpression of NLRC5 affected the

expression levels of VSMC proliferation and dedifferentiation markers under basal conditions without PDGF-BB stimulation.

On page 19,

Consistent with the *in vivo* findings, *NLRC5* knockdown reduced the expression of α -SMA, Calponin, and Myosin concomitant with increased expression of proliferative markers PCNA and Cyclin D1 under basal conditions and after PDGF-BB stimulation. These data indicate that *NLRC5* depletion promoted VSMC dedifferentiation, a process of switching from a contractile to proliferative phenotype (Figure 3E and 3F).

7. Nuclear localization of NLRC5 in SMCs was overinterpreted and incorrect. In *in vivo* images (Figure 1B&D, Sup Figure 2) and *in vitro* images (Figures 1J, 4B&D) there is clearly cytoplasmic localization (there would be no yellow staining in overlay images with NLRC5 and α SMA staining if NLRC5 was strictly nuclear as α SMA is restricted to the cytoplasm). Please alter the text to reflect this (in particular lines 568-569).

Reply: We acknowledge this point and have updated the relevant sentences as suggested.

On page 15,

Notably, *Nlrc5* expression was abundant in the neointima rather than the media in the injured artery, and was located predominantly more in the nucleus rather than in the cytoplasm of α -smooth muscle actin (α -SMA) positive VSMCs.

On page 15,

NLRC5 expression gradually increased in the nucleus and reached its peak at 6 hours following PDGF-BB stimulation (10 ng/ml), which was maintained up to 12 hours. In

contrast, the expression of NLRC5 in the cytoplasm of HASMCs was at a very low level and remained unchanged following PDGF-BB stimulation (10 ng/ml) (Figure 1I). Although there was rare cytoplasmic localization, NLRC5 was predominantly expressed in the nuclei of HASMCs and increased in response to PDGF-BB treatment (10 ng/ml) (Figure 1J).

8. The number of human samples analyzed is very low and only N=1 from normal vs. atherosclerotic was shown. The representative atherosclerotic images in the immunofluorescent images is not representative as there is very little intimal hyperplasia (this image is more representative of a normal human coronary vessel). Please include example confocal images from all stained human vessels in a supplementary figure. In addition, as the authors referred to “the proliferated medial layer...” (line 293), triple staining for NLRC5, α SMA, and PCNA would strengthen the data.

Reply: We thank the reviewer for these helpful points. As suggested, we have replaced a pair of representative immunofluorescent images of normal coronary and coronary plaques stained with NLRC5 and α -SMA. Additionally, we have presented all the samples with immunofluorescence staining in the supplementary figure. However, we should acknowledge that we had attempted to perform triple co-staining of NLRC5, α -SMA, and PCNA but failed. As an alternative approach, we performed immunohistochemistry staining with NLRC5, α -SMA, and PCNA in serial sections of human coronary plaques, and found that NLRC5 was ubiquitously expressed in the coronary in addition to the proliferative medial layer. The corresponding data had been rewritten and the corresponding figures had been updated.

On page 14,

Compared with normal coronary arteries, NLRC5 expression was more abundant in VSMCs in both Kawasaki disease (Supplementary Figure I) and coronary plaques; however, its expression was also ubiquitous, rather than localized in the proliferative medial layer (Figure 1A-C and Supplementary Figure II).

Figure 1B

Supplementary Figure II

A

B

Minor

9. p.20, lines 439-444: the text is confusing as the authors first refer to pioglitazone stimulation, then change to PDGF-BB stimulation. Please address and correct.

Reply: We acknowledge this error and have incorporated these changes in these sentences.

On page 24,

The ChIP-qPCR results showed that PPAR γ -binding to the NLRC5 promoter significantly increased in response to pioglitazone stimulation (Figure 5H), further supporting our hypothesis.

10. Figure 5 and text: Please address the significance and define the various NLRC5 domains.

Reply: We have now incorporated explanations for the definition and significance of the various NLRC5 domains in the text and figure legend.

On page 25,

NLRC5 consists of a tripartite structure, including an N-terminus, a central NACHT domain, and a C-terminal leucine-rich repeat domain (LRR). Unlike most of NLR family members, the N-terminus of NLRC5 possesses an atypical caspase activation and recruitment domain (CARD), also known as death-domain-like fold (DD). On the basis of the structural database information (UniProtKB-Q86WI3, NLRC5_HUMAN, <https://www.uniprot.org/uniprot/Q86WI3#structure>), we defined the DD domain of NLRC5 as amino acid (aa) 1-221 and the NACHT domain as aa 222-539.

On page 32,

Human NLRC5, the largest member in the NLR family, is composed of DD domain, NACHT domain and LRR domain, and exhibits structural homolgy to CIITA^{15, 32}. Previous studies revealed that the DD domain conferred transcriptional activity and regulated MHC-I/HLA transcription. The NACHT domain is known as the nucleotide-binding domain thought to be critical for nuclear import and transactivation activity,¹⁹ whereas the LRR domain is responsible for ligand binding.

Figure 6A. Schematic diagram of four *NLRC5* mutant constructs. The structural domains are defined according to the structural database information (UniProtKB-Q86WI3, NLRC5_HUMAN, <https://www.uniprot.org/uniprot/Q86WI3#structure>).

11. Figure 2: Please show higher magnification images of PCNA-stained arteries to better appreciate the data (similar to Figure 6F).

Reply: We have presented the representative images of PCNA staining at higher magnification in Figure 2E and 2K.

12. Figure 4A: Please show reciprocal IP (IP NLRC5, IB PPAR γ).

Reply: We have performed the reciprocal IP as suggested and presented the data in Figure 4A.

On page 21,

PDGF-BB stimulation (10 ng/ml) remarkably promoted the intrinsic interaction of NLRC5 with PPAR γ in HASMCs within 6-12 hours (Figure 4A). Vice versa, immunoprecipitation with NLRC5 antibodies followed by immunoblot analyses showed an increase in the interaction of NLRC5 with PPAR γ in the presence of PDGF-BB (Figure 4A).

13. Figure 4F and Figure 5E-G: It is difficult to appreciate a dose response of overexpression making the robust response at the highest dose questionable. Please address.

Reply: As suggested by the reviewer, we repeated the experiments with additional time points and updated the results.

On page 25,

Moreover, compatible with increased protein expression, induction of NACHT, NACHT+DD, and ISO3 remarkably enhanced PPAR γ activity and reached their

peaks at 2 or 3 μg concentrations, while DD mutant did not affect PPAR γ activity (Figure 6E-H).

14. More detail should be included in the figure legends (e.g. number of independent experiments/westerns conducted; quantification of NLRC5-, PCNA-positive cells)

Reply: As suggested, we now provided additional details in the figure legends.

15. Are the data in Sup Figure IV significant for CD8⁺ T-cells? If so, please indicate on the figure.

Reply: Thank you for your suggestion. We had repeated the statistical analysis and found significant reductions of the percentage of CD8⁺ T cells in spleen and peripheral blood from *Nlrc5*^{-/-} mice as compared with those from *Nlrc5*^{+/+} mice. The asterisks had been marked in Supplementary Figure IV.

On page 17,

Consistent with previous reports, flow cytometric analysis showed a significant reduction in the percentage of CD8⁺ T cells in splenocytes (*Nlrc5*^{+/+} 10.7 \pm 1.2% vs. *Nlrc5*^{-/-} 5.7 \pm 0.6%, P = 0.010, Supplementary Figure VIA), and in peripheral blood

(*Nlrc5*^{+/+} 8.8 ± 0.3% vs. *Nlrc5*^{-/-} 7.2 ± 0.3%, P = 0.011, Supplementary Figure VIB) of *Nlrc5*^{-/-} mice.

REVIEWERS' COMMENTS:

Reviewer #1 (Remarks to the Author):

I applaud the authors for the way they responded to the reviewers comments constructively and with additional data. You are to be commended.

Reviewer #2 (Remarks to the Author):

The manuscript has been greatly improved and shall be accepted for publication. One important aspect that shall however be clearer in title, abstract, text, and discussion is that - even if there are strong biochemical/overactivation data supporting the involvement of PPAR γ in NLRC5 phenotype in the neointima - we do not have endogenous/functional data directly implying PPAR γ in the NLRC5 phenotype. I therefore propose a title like "NLRC5 inhibits neointima formation following vascular injury and directly interacts with PPAR γ in this context", and corrections along the same line throughout the text.

Reviewer #3 (Remarks to the Author):

The authors have addressed most of my previous concerns and have added new data to support the overall hypothesis and findings. There are some issues that need to be addressed, as outlined below.

Major:

1. There are several instances of overstated and/or overinterpreted data:
 - The authors continue to state that NLRC5 is exclusively nuclear in SMCs. However, the immunofluorescence data simply do not support this. There is considerable co-expression of NLRC5 and α SMA (yellow fluorescence) in the images, human and mouse tissues. As α SMA is exclusively cytoplasmic, there is clearly an abundance of NLRC5 localized to the cytoplasm. It is recommended that the authors reword their findings throughout the manuscript for accuracy.

- Line 336: “expression gradually increased in the nucleus and reached its peak at 6 hours...” The authors only looked at 0 and 6 hours, no earlier time points, therefore results can only state that nuclear NLRC5 expression was observed at 6 hours and was maintained up to 12 hours.
- Interpretation of the data related to the siRNA studies should be downplayed (i.e. “These experiments indicated that 482 NLRC5 predominantly interacted with PPAR γ , acted as a PPAR γ ligand and regulates PPRE-483 dependent gene transcription without alteration of the PPAR γ /RXR α complex.”). As NLRC5 (and PPAR γ) was only partially silenced and significant residual levels of protein remain, this residual level could still be affecting PPAR γ /RXR α .
- The data suggesting that pioglitazone enhances NLRC5-PPAR γ interactions is not convincing. Importantly, the authors demonstrate that NLRC5 interacts with the ligand binding domain of PPAR γ , which is also the domain of pioglitazone-PPAR γ interaction. The authors should discuss this in more detail, in particular why there wouldn’t be competition between pioglitazone and NLRC5 binding.

2. Lines 402 and 403. “Myosin, α -SMA, and Calponin have previously been 402 reported as key regulators of downstream VSMC differentiation gene expression.” Please reword this statement. As addressed previously, myosin, α SMA, and calponin do not regulate SMC differentiation. They are part of the SMC contractile machinery and define a differentiated SMC.

Minor:

1. Line 394. This should be supplementary figure X.
2. Supplementary figure IV. Why is there positive NLRC5 staining in NLRC5 WT mice where no such expression was observed in Figure 1?
3. Please remove line 520 as this is redundant with line 519.
4. Line 573. Shouldn’t this read “markedly decreased?”
5. Line 505. “deterioration of PCNA and ...” It is unclear what the authors mean here. T007 reversed NLRC5-mediated downregulation of PCNA and cyclin D1.
6. Figure 4C. Please show double immunofluorescent confocal images of NLRC5 and PPAR γ if stating these are co-localized. Serial sections cannot be used to adequately demonstrate co-localization, only a similar pattern of staining.
7. Figure 6K. These data might be consistent with PPAR γ -mediated transcriptional activity, but are not consistent with PPAR γ /NLRC5-mediated inhibition of SMC proliferation and migration as these target genes have been shown to enhance SMC proliferation. (for instance, please see PMC6345504). This should be discussed.

Response to reviewers

Reviewer #1 (Remarks to the Author):

I applaud the authors for the way they responded to the reviewers comments constructively and with additional data. You are to be commended.

Reply: Thank you for your comments and we greatly appreciate your hard work and constructive suggestions.

Reviewer #2 (Remarks to the Author):

The manuscript has been greatly improved and shall be accepted for publication. One important aspect that shall however be clearer in title, abstract, text, and discussion is that-even if there are strong biochemical/overactivation data supporting the involvement of PPAR γ in NLRC5 phenotype in the neointima - we do not have endogenous/functional data directly implying PPAR γ in the NLRC5 phenotype. I therefore propose a title like "NLRC5 inhibits neointima formation following vascular injury and directly interacts with PPAR γ in this context", and corrections along the same line throughout the text.

Reply: We highly appreciate the reviewer's suggestion. We acknowledge these weaknesses in our study and now rephrase the words in the corresponding sentences throughout the text.

Reviewer #3 (Remarks to the Author):

The authors have addressed most of my previous concerns and have added new data to support the overall hypothesis and findings. There are some issues that need to be addressed, as outlined below.

Reply: We thank the reviewer and correct all issues raised by the reviewer.

Major:

1. There are several instances of overstated and/or overinterpreted data:

The authors continue to state that NLRC5 is exclusively nuclear in SMCs. However, the immunofluorescence data simply do not support this. There is considerable co-expression of NLRC5 and α SMA (yellow fluorescence) in the images, human and mouse tissues. As α SMA is exclusively cytoplasmic, there is clearly an abundance of NLRC5 localized to the cytoplasm. It is recommended that the authors reword their findings throughout the manuscript for accuracy.

Reply: We acknowledge these errors and rephrase the sentences as suggested.

For example, on page 19,

Interestingly, our current study shows that NLRC5 protein is highly expressed in the nucleus of stimulated VSMCs.

2. Line 336: "expression gradually increased in the nucleus and reached its peak at 6 hours..." The authors only looked at 0 and 6 hours, no earlier time points, therefore results can only state that nuclear NLRC5 expression was observed at 6 hours and was

maintained up to 12 hours.

Reply: We agree with the reviewer and rephrase the sentence as suggested.

On page 5,

NLRC5 increased at 6 hours in the nucleus and was maintained up to 12 hours following PDGF-BB stimulation (10 ng/ml).

3. Interpretation of the data related to the siRNA studies should be downplayed (i.e. “These experiments indicated that 482 NLRC5 predominantly interacted with PPAR γ , acted as a PPAR γ ligand and regulates PPRE-483 dependent gene transcription without alteration of the PPAR γ /RXR α complex.”). As NLRC5 (and PPAR γ) was only partially silenced and significant residual levels of protein remain, this residual level could still be affecting PPAR γ /RXR α .

Reply: Indeed, we could not entirely exclude the possibility of the residual effect of NLRC5 on PPAR γ /RXR α complex via these proposed methods in our work. We now rephrase the sentence.

On page 12,

These experiments indicate that NLRC5 predominantly interacts with PPAR γ , acts as a PPAR γ ligand and regulates PPRE-dependent gene transcription. Indeed, because of the residual levels of NLRC5 protein after knockdown of NLRC5, we could not entirely exclude the potential effect of NLRC5 on PPAR γ /RXR α complex in HASMCs.

4. The data suggesting that pioglitazone enhances NLRC5-PPAR γ interactions is not convincing. Importantly, the authors demonstrate that NLRC5 interacts with the ligand binding domain of PPAR γ , which is also the domain of pioglitazone-PPAR γ interaction. The authors should discuss this in more detail, in particular why there wouldn't be competition between pioglitazone and NLRC5 binding.

Reply: We acknowledge these errors and overinterpretation. Our data coupled with previous literatures suggested that pioglitazone not only increased PPAR γ expression but also induced high-affinity interaction with PPAR γ -LBD domain. Furthermore, in Figure 5C-5E, we found that pioglitazone promoted NLRC5 expression via recruitment of PPAR γ at the promoter of NLRC5 and activation of NLRC5 transcription. In fact, since the increased combination of NLRC5 and PPAR γ paralleled the elevated expression of NLRC5 induced by pioglitazone, we could not conclude whether exogenous PPAR γ ligand, pioglitazone, enhanced or competed with NLRC5-PPAR γ interaction.

On page 13,

Notably, the intrinsic interaction between NLRC5 and PPAR γ existed without stimulation of the exogenous PPAR γ ligand (Figure 5E). However, since the increased combination of NLRC5 and PPAR γ paralleled the elevated expression of NLRC5 induced by pioglitazone, we could not conclude whether exogenous PPAR γ ligand,

pioglitazone, enhanced or competed with NLRC5-PPAR γ interaction.

5. Lines 402 and 403. “Myosin, α -SMA, and Calponin have previously been reported as key regulators of downstream VSMC differentiation gene expression.” Please reword this statement. As addressed previously, myosin, α -SMA, and calponin do not regulate SMC differentiation. They are part of the SMC contractile machinery and define a differentiated SMC.

Reply: We are sorry for these inappropriate definitions and rephrase them now.

On page,

Myosin, α -SMA, and Calponin are considered as VSMC differentiation markers.

Minor:

1. Line 394. This should be supplementary figure X.

Reply: We acknowledge this error and replace the annotation.

On page 8,

We found that NLRC5 overexpression prominently alleviated PDGF-BB-induced HASMCs proliferation (Supplementary Figure 10A).

2. Supplementary figure IV. Why is there positive NLRC5 staining in NLRC5 WT mice where no such expression was observed in Figure 1?

Reply: Indeed, *Nlrc5* is moderately expressed in vessels of WT mice (stained in red) as presented in Figure 1D. It is worth noting that the specificity of *Nlrc5* staining was verified in ligated carotid arteries of WT mice (stained in green) shown in Supplementary Figure IV. We now intercept these two images and clarify *Nlrc5* expression in vessels. Furthermore, to make our data clear, we emphasize the expression of *Nlrc5* and samples in the text and figure legends.

On page 6,

*We verified the success of *Nlrc5* deletion in *Nlrc5*^{-/-} mice and tested the specificity of *Nlrc5* staining in ligated carotid arteries (Supplementary Figure 4).*

Figure 1D

Supplementary Figure IV

3. Please remove line 520 as this is redundant with line 519.

Reply: We are sorry for this mistake and remove the repetitive sentence.

4. Line 573. Shouldn't this read "markedly decreased?"

Reply: We have corrected the expression as follows.

On page 15,

*In accordance with the in vitro observations, the cross-sectional areas of neointima (Ad-Control $1.29 \pm 0.38 \times 10^4 \mu\text{m}^2$ vs. Ad-Nlrc5 NACHT $0.48 \pm 0.17 \times 10^4 \mu\text{m}^2$, $P < 0.01$ by Student's *t*-test) and intima/media ratios (Ad-Control 0.60 ± 0.11 vs. Ad-Nlrc5 NACHT 0.32 ± 0.07 , $P < 0.01$ by Student's *t*-test) in ligated carotid arteries were markedly decreased in Ad-Nlrc5 NACHT mice compared with Ad-Control mice (Figure 6M-O).*

5. Line 505. "deterioration of PCNA and ..." It is unclear what the authors mean here. T007 reversed NLRC5-mediated downregulation of PCNA and cyclin D1.

Reply: As suggested, we rewrite this sentence.

On page 13,

Conversely, the enhancement of α -SMA, Calponin, and Myosin expression in HASMCs transduced with Ad-NLRC5 was blocked in the presence of the T0070907, and accompanied by increases in PCNA and Cyclin D1 expression.

6. Figure 4C. Please show double immunofluorescent confocal images of NLRC5 and PPARgamma if stating these are co-localized. Serial sections cannot be used to adequately demonstrate co-localization, only a similar pattern of staining.

Reply: We agree with the reviewer and design the double immunofluorescence staining. We could identify the co-localization of Nlrc5 and PPAR γ through double immunofluorescence staining in Figure 4C.

On page 11,

In parallel, we found that Nlrc5 expression co-localized with PPAR γ in mouse carotid arteries after 1-week of carotid ligation through double immunofluorescent staining (Figure 4C).

7. Figure 6K. These data might be consistent with PPAR γ -mediated transcriptional activity, but are not consistent with PPAR γ /NLRC5-mediated inhibition of SMC proliferation and migration as these target genes have been shown to enhance SMC proliferation. (for instance, please see PMC6345504). This should be discussed.

Reply: We appreciate the reviewer's critical comments. We have reviewed the relevant literatures, finding that activation of PPAR γ protects vascular hyperplasia and atherosclerosis. However, some of PPAR γ target genes, such as CD36, promote VSMC proliferation and aggravate neointimal formation. In other studies, PPAR γ agonists pioglitazone increases the incident risk of bladder cancer. Therefore, it could be explained that while activation of PPAR γ retards arteriosclerotic cardiovascular disease and type 2 diabetes, some of PPAR γ target genes result in unfavorable effects on diverse tissues. Due to these board side effects, the physicians become more cautious when prescribing PPAR γ agonists in clinics.

On page 21,

While PPAR γ agonists and most of PPAR γ target genes are proven protection in type 2 diabetes and cardiovascular diseases, a minority of PPAR γ target genes, such as CD36, are reported to promote VSMC proliferation and tumor metastasis. Given the side effect issues associated with PPAR γ agonists, including tumor formation, bone fractures, weight gain, and fluid retention, targeting NLRC5 may provide a valuable means to modulate PPAR γ signaling and potentially overcome these issues.